



# Synoptic and Mesoscale atmospheric features associated with an extreme Snowstorm over the Central Andes in August 2013

Marcelo Zamuriano[1], Paul Froidevaux[1,2], Isabel Moreno[3], Mathias Vuille[4], and Stefan Brönnimann[1]

[1]Oeschger Centre for Climate Change Research and Institute of Geography, University of Bern, Switzerland
[2]Meteotest AG, Bern, Switzerland
[3]UMSA/LFA, La Paz, Bolivia
[4]Dept. of Atmospheric and Environmental Sciences, Univ. at Albany, USA

**Correspondence:** Marcelo Zamuriano (marcelo.zamuriano@giub.unibe.ch)

**Abstract.**

We study the synoptic and mesoscale characteristics of a snowfall event over the Bolivian Altiplano in August 2013 that caused severe damage to people, infrastructure and livestock. This event was associated with a cold front episode following the eastern slope of the Andes-Amazon interface and a cut-off low pressure system (COL) over the Pacific Ocean. Large scale

analyses suggest a two-stage mechanism: The first phase consisted of a strong cold surge to the east of the Andes inducing low level blocking of southward moisture transport over the SW Amazon basin due to post-frontal high-pressure up to 500 hPa synchronized to a Rossby wave train. The second stage was initiated by the displacement of 500 hPa anticyclone over the Andes due to a Rossby wave passage and a subsequent increase in north-easterly moisture transport, while another cold front along the eastern Andes provided additional lifting. We analyse an analog event (July 2010) to confirm the influence of these

large-scale features on snow formation.

Furthermore, we conduct a mesoscale analysis of the 2013's event using the Weather Research and Forecasting (WRF-ARW) model. For this purpose, we perform a series of high-resolution numerical experiments that include sensitivity studies where we apply orographic and lake Titicaca temperature modifications. We compare our findings to MODIS snow cover estimates and in-situ measurements. The control simulation is able to capture the snow cover spatial distribution and sheds light over

several aspects of the snowfall dynamics. In our WRF simulations, daytime snowfall mainly occurs around complex orography whereas nocturnal snowfall is concentrated over the plateau due to a combination of nocturnal winds and complex orography inside the plateau. The sensitivity experiments indicate the importance of the lake and mountain for thermal wind circulation affecting the spatial distribution of snowfall by shifting the position of the convergence zones. The influence of the lake's thermal effect is not evident around the regions surrounding the lake.

# 1 Introduction

Over the flat and wide region of the central tropical Andes, also known as Altiplano, about 70% of the yearly precipitation occurs from December to March (Garreaud et al., 2003b; Andrade (Ed.), 2018). The wintertime season (June, July and August, hereafter JJA) is characteristically dry in response to an intensification of westerly dry flow from the Pacific (Vuille, 1999;



Garreaud, 2009). This dry flow can occasionally be suppressed by low-level cold surges episodes from the south; destabilizing the atmosphere and resulting in convective precipitation over the Amazon (Garreaud and Garreaud, 1999; Espinoza et al., 2012, 2015).

Winter precipitation over the tropical Altiplano is usually beneficial for multiple uses. It reduces the glaciers' melt rate (Sicart et al., 2011); it regulates the disease control affecting agriculture (UNDP, 2011); it provides water for agricultural purposes (Helvetas, 2014); and it is even used as a proxy by local farmers for predicting the rainy season's "quality" (Boillat and Berkes, 2013). However, extreme snowfall is considered one of the most hazardous meteorological phenomena. It can block roads, cause airport closures and affect farmer's activities.

Over the Tropical Andean region, snowfall episodes can be related to upper-level cut-off low (COL) and cold frontal events (Vuille and Ammann, 1997). Sicart et al. (2015) have found that around 87% of cloudy days over the Bolivian glacier Zongo (16°15'S, 68°10'W) in JJA, from 2005 to 2013, occurred during a cold surge passage; while the remaining 13% of cloudy days were associated with a high-level low pressure conditions (including COLs) along the Chilean cost around 45°S.

Large-scale atmospheric circulation related to wintertime cold surges, along the eastern slope of the Andes, have been extensively studied due to the dramatic consequences of frosts on crop production and daily life (Pezza and Ambrizzi, 2005; Sprenger et al., 2012; Lanfredi and de Camargo, 2018; Mintegui et al., 2018). Cold surges mainly follow three south-north trajectories: the first one along the Atlantic coast, the second one following the Parana river, and the third one to the east of the Andes (Lupo et al., 2001). Cold air incursions along the eastern slope of the Andes have been less studied than the other trajectories, but recent efforts have tried to fill that gap, providing descriptions of the propagation characteristics (Espinoza et al., 2013), their influence on cloudiness (Sicart et al., 2015) and precipitation over the eastern part of the Altiplano (Perry et al., 2017; Hurley et al., 2015).

Snowfall episodes related to upper-level lows have typically been studied along the Pacific coast and most studies are focused on extra-tropical regions. Such episodes can be accompanied by south-north propagating cold fronts over the Pacific ocean and the effect of the Andes on the spatial distribution of snowfall is clear (Garreaud and Fuenzalida, 2007; Viale et al., 2011; Scaff et al., 2017). Nevertheless, similar studies are still missing over the tropical Andes' region.

One particular strong snowfall event over the Altiplano that included both a COL formation over the Pacific coast and a cold front over the Andes eastern slope was observed on August 2013, during a particularly cold winter in South America. From 23 to 25 August 2013, a heavy snowfall episode killed over 26,600 animals, left over 5,250 people homeless, and destroyed nearly 140 homes in the Peruvian Andes (PNUD, 2013). In the Bolivian Andes, the same event left 3 casualties, around 3,500 affected families and 20,915 dead animals from livestock and an emergency state was declared (de Defensa de Bolivia, 2013). Northern Chile also registered a historical snowfall accompanied with strong wind over the Atacama desert (ONEMI, 2013).

Taking into account the range of impacts that snowfall can have over the tropical central Andes, it is fair to say that an accurate prediction of the timing, intensity and spatial distribution is crucial. Several efforts have been made in this direction, ranging from large-scale atmospheric circulation (Sicart et al., 2015; Vuille et al., 2008; Espinoza et al., 2013), to mesoscale circulation induced by orographic effects (Zängl and Egger, 2005; Junquas et al., 2017; Mourre et al., 2016), and precipitation





structure (Perry et al., 2017; Comin et al., 2018). However, many aspects of winter precipitation over this region are still poorly understood.

This study aims to explore the complex interactions between large-scale circulation and orographic features leading to heavy snowfall episodes over the central Andes. For this purpose, we set two specific goals. First, we study the main characteristics

of the 2013 event from a synoptic perspective (using ERA-interim reanalysis) and from a mesoscale circulation view by performing numerical simulations. The second goal is to assess the importance of regional features by conducting a series of sensitivity experiments, including orography modification and lake removal.

The organization of the paper is as follows. We introduce the datasets along with the methods used, including the numerical simulations set-up, in section 2. Section 3 describes the main characteristics of the 2013's event (we also analyse an analog

case for comparing the synoptic circulation), including numerical experiments results. Section 4 contain the discussion of the main findings and section 5 summarize the main results and include the conclusions of this study.

## 2 Data and methods

### 2.1 Datasets

The nature of the study area demands adequate datasets and analysis methods, which unfortunately are not always available

over the study area. For the following study about heavy snowfall over complex orography we used the next datasets.

#### 2.1.1 Surface Data

Recent studies have evaluated the quality of precipitation measurements over the Central Andes(Hunziker et al., 2017, 2018). Hunziker et al. (2017) remarked that in many stations, no observations are taken on a certain day of the week, mostly for regional airports during the weekend. Since the 2013 event took place during a weekend, we note that some stations may not

have trustful data available, which are duly flagged in the dataset produced the 'Data on climate and Extreme weather for the Central Andes' project (DECADE). The dataset is based on measurements made by the Bolivian and Peruvian National Weather Services (hereafter SENAMHI) and it is available on http://www.geography.unibe.ch/research/climatology_group/ research_projects/decade/index_eng.html. The DECADE dataset includes daily measurements of total precipitation from 235 stations of which 184 are located higher than 1500 meters above sea level.

We complement the station information with national civil protection reports and documents produced by international organizations such as the United Nations UNDP (2011); (SENAMHI); Helvetas (2014); PNUD (2013); ONEMI (2013). These reports provide quantitative and qualitative information about the impact of the event in places where no measurements are available.



### 2.1.2 Remote sensing data

Snow cover data is obtained from the MODIS Aqua satellite daily snow cover dataset MYD10C1 (Hall and Riggs, 2016). Because the reflectance of fresh snow is high, this product is able to capture the spatial distribution of snow cover over land under the condition of a clear sky. This conditions is fulfilled after the event and therefore we take the data corresponding

to 25th August 2013. TRMM-based precipitation estimates from TMPA 3B42RT are ruled out since this product does not adequately capture snowfall (Beck et al., 2017)

In addition to MODIS, we use the Geostationary Operational Environmental Satellite (GOES-12 and 13) in the water vapour spectral and visible bands (6.48 and 0.63 μm respectively) for an spatial assessment about cloud cover.

### 2.1.3 Global analysis

To asses large-scale circulation and moisture transport, we extract fields from the ERA-interim global reanalysis (Dee et al., 2011) produced by the ECMWF. This product has temporal outputs available each 6 hours with a grid resolution of $0.75° \times$ 0.75 ° lat-lon; the vertical axis contains 60 levels from surface up to 0.1 hPa pressure level.

Even though we use ERA-interim for large-scale circulation and climatology assessment, we decided to use the Final Analysis (FNL) from the Global Forecast System (GFS) as boundary and initial conditions for the WRF simulations. This choice

is made because of the better reproduction of spatial snow-cover distribution by FNL-based simulations with respect to ERA-interim based simulations (not shown). Additionally, we note SENAMHI's GFS products utilization for operational forecasts. Hence they may be able to directly profit from our study.

### 2.2 Methods

#### 2.2.1 Meteorological diagnostics

One goal of this study is to analyse the influence of cold surges' position along the Andes on snowfall episodes. There exist several front detection methods, but assessing which method is the most accurate over this region is beyond the scope of this study. Our goal is to focus on the position of the front and with this in mind, we decided to use the equivalent potential temperature $\theta_e$ gradient at 850 hPa because of its simplicity and its consistency with synoptic features in moist processes(Schemm et al., 2018)

Following Sprenger et al. (2012), we aim to relate the cold surges to polar and upper-level influences and therefore we decide to examine the upper-level conditions through a potential vorticity (PV) perspective in PVU units.

We complement the cold surge study with an assessment of large-scale moisture transport, with the calculation of the vertically-Integrated water Vapour Transport (IVT) vector,





$$IVTU = \frac{1}{g} \int_{SFC}^{200} qu \, dp \qquad (1)$$

$$IVTV = \frac{1}{g} \int_{SFC}^{200} qv \, dp \qquad (2)$$

where $q$ is the specific humidity ($\mathrm{gkg^{-1}}$), $g = 9.81$ $\mathrm{ms^{-2}}$ is the gravity constant, $u$ and $v$ are the zonal and meridional wind in $\mathrm{ms^{-1}}$ and $dp$ is the pressure thickness (hPa). We calculate it starting from surface $SFC$ to the 200 hPa level.

The circulation of cold dry air within the cold surges and the trajectories of moist parcels is studied with a Lagrangian trajectory diagnostic. We use the LAGRANTO tool (Sprenger and Wernli, 2015) for backward and forward trajectory of air masses off-line calculation in WRF outputs.

The mesoscale analysis is performed by studying several diagnosis such as divergence, relative humidity and cross sections over selected areas.

### 2.2.2 Numerical model set-up

Mesoscale characteristics are investigated through dynamical downscaling with the Advanced Research (WRF-ARW) core of the Weather Research and Forecasting Model version 3.9.1.1 (Skamarock et al., 2008). We define three two-way nested domains (i.e. with feedback from inner to outer domains), Fig. 1a) named D1, D2 and D3, with a horizontal resolution of 27, 9 and 3 km respectively and 60 vertical levels up to 50 hPa.

The simulations are initialised on 21 August 2013 at 0000 UTC and runs until after the event. The main configuration is summarized in table 1 and it is based on: the Thompson microphysics scheme (Thompson et al., 2008), the Yonsei University (YSU) planetary boundary layer scheme (Hong et al., 2006) with orographic effect corrections in the finest domain (Jiménez et al., 2012), the Rapid Radiative Transfer Model (Mlawer et al., 1997) for long and short-wave radiation, and the Noah-MP land surface model (Niu et al., 2011). We use the Kain-Fritsch scheme (Kain, 2004) for cumulus parametrization only for domains D1 and D2; while D3 uses a explicit cumulus treatment. We also use spectral nudging (von Storch et al., 2000) in order to allow WRF to follow the large scale circulation in domain D1 above the planetary boundary layer; relaxing slightly the large-scale structure of temperature, geopotential and wind (wave-lenghts larger than 1000 km) towards the input GFS FNL data. Humidity fields are not nudged. The simulation with the model's unchanged orography and realistic lake temperature is set as the control run (CTRL) and it is used for the mesoscale features assessment.

Following Zamuriano et al. (2019), in addition to the CTRL experiment, we study the impact of the Andes' orography by reducing the peaks' elevation above the Altiplano (3750 masl) by half (experiment RTA), and by reducing orography to 80 % of its original height over all domains (experiment RT80). We also study the influence of lake Titicaca by reducing its surface temperature by 5 °C (experiment LK-5), and by replacing the lake it by the land category around it (we call it experiment NOLAKE). A description of the experiments with the main differences are described in table 2


## 3   Results

### 3.1   Snowfall chronology

#### 3.1.1   Synoptic circulation

The heavy snowfall episode over the Andes occurred between the afternoon of 23 August and the morning of 25 August 2013.

The Bolivian Civil protection office reported more than 1 meter of snow height in Tinguipaya (hereafter TP) and Cocapata (hereafter CP), the two most affected regions (de Defensa de Bolivia, 2013).

Brightness temperature derived from GOES-13 satellite in the water vapour spectral band (6.48 μm) suggests cloud formation over the north-western part of the Amazon (northern Bolivia) and over the south-eastern Pacific ocean two days prior the first event (22 August 0600 at UTC, Fig. 1b). During the first night of snowfall (24 August 0600 UTC), a large amount of cloud

cover was present all over northern Bolivia, the Altiplano and south-eastern Bolivia (Fig. 1c).

The evolution of the synoptic conditions from 21 to 26 August (Figs. 2a-e) shows that both stages of cloud cover in Figures 1b-c are interconnected by a large amount of total column water displacement from the north towards the Altiplano, accompanied by the appearance of a quasi-stationary cold-front along the Andes (Fig. 2c-e).

An analysis of the low mid-level circulation shows the establishment of an anticyclone at 500 hPa (we stress that this level

correspond to around 1500 meters above surface over the Altiplano) on 21 August over the north-west Altiplano (Fig. 2a) after the passage of a previous strong cold front (on 15 August, discussed in the next paragraph). This anticyclone is accompanied by high amount of precipitable water northern Bolivia seen in Fig. 2a-c. The daily mean of precipitable water captured by ERA-interim over the blue box in Fig. 2a exceeds the 85 percentile for JJA during the full analysis period with some deformations showing moisture transport southwards (Fig 2c-e).

A closer look to the upper troposphere (isentropic level corresponding to 330 deg C, equivalent to around 250 hPa) and cold fronts position (characterized by high sea level pressure) show that both cold fronts of 15 and 23 August were in phase with the Rossby wave trains (Figs. 3a and 3e), portrayed by PVU lower than -2 units. While this happened, over the Pacific ocean, the Rossby waves reached the critical level leading to wave breaking (Figs. 3b-d). This generated a trough that was later reinforced by the next wave and formed a cut-off low system over the western Altiplano (Figs. 3e-f).

IVT analysis further indicates that the moisture is transported southwards from the north-west Amazon towards and over the Altiplano following the Andes slopes. It shows also some signals over the Pacific ocean by occasions (Fig. 4a-d). At the same time, IVT shows that once reaching the Altiplano, moist air is transported to the south-east following the position of the continental cold front. A climatological study of IVT over two selected grid points located along the Andes-Amazon transition (Points A1 and A2 in Fig. 4a-d) indicates that the moisture transport during this event was extreme for this region; the polar

diagrams (Fig. 4e and 4f) reveal that IVT originating out of the north-west surpassed even the 99 percentile for all seasons.





## 3.2 Analog snowstorm: the 17 July 2010 case

In order to confirm that the main synoptic features associated with the 23-25 August 2013 snowstorm are characteristic of similar events leading to winter snowfall in this region, we analyse a 2nd analog event which occurred during a cold surge on 17 July 2010. This event was considered by Lanfredi and de Camargo (2018) as the twenty-first century's coldest air incursion

along the eastern slope of the Andes, and it features many similarities with the 2013 event.

An upper-level COL developed over the southwest Altiplano and a quasi-stationary cold front was present to the east of the Andes (Fig 5a,d,e). At the same time, southward vapour transport towards the Altiplano intensified (Fig. 5b) once the mid-level high-pressure system was displaced by appearance of the COL (Fig. 5f). The main event occurred between 17 and 18 July and it was located more to the east, following the COL position (Fig. 5c).

### 3.2.1 Snow cover evolution and station measurements

The large-scale circulation analysis is able to explain the large amounts of snow that affected the Altiplano but offers little detail about the spatio-temporal distribution. A first assessment of the snow cover evolution is made with the MODIS MYD10C1 daily snow cover product on 25 August (once the event is over and the reduced cloudiness allows a reasonable snow cover estimate) to validate the snowfall results from the control run.

The snow-covered areas on 25 August (Fig. 6a) were mainly located over complex orography with a particular concentration over higher elevations of the eastern and western cordillera, encompassing the Altiplano. The snow-covered area over the interior of the southern Altiplano, is likely misclassified by MODIS as it closely follows the lowest depressions of the highly reflective salt flats of Uyuni and Coipasa, which are highly unlikely to have snow cover while the surrounding higher elevations are bare of snow. MODIS captures the snow cover over the most affected areas (CP and TP), and over the most populated cities

(La Paz and Oruro).

As seen in Figure 6b, the WRF CTRL simulation is capable to emulate the timing and spatial distribution of snow cover but not the alleged intensity (The Bolivian Civil protection agency reported up to three meters of snow, while WRF simulates at most one meter). The reasons of this underestimation can range from the choice of the microphysics schemes, the land use models and the accuracy of the reports. Nevertheless, it is possible to numerically investigate the mesoscale circulation and the

main dynamical mechanisms that may have led to this historical event.

Additional SENAMHI measurements of daily precipitation (we remind that measurements are taken from 1200 UTC every 24h) show a clustering around La Paz city from 23 to 24 August (Fig. 6c) and a more homogeneous distribution from 24 to 25 August (Fig. 6d). The lack of stations in southern Altiplano or even southern Oruro is evident.

## 3.3 Mesoscale dynamics from control run

In the following section we analyse daytime and nocturnal snowfall separately. Taking into consideration the region time zone (UTC-4), we consider as night the period from 0000 to 1200 UTC and as day the period from 1200 to 0000 UTC. Daytime





snowfall is mostly concentrated over complex orography (Figs. 7a,c), while night-time snowfall is more homogeneous and extends over much of the Altiplano, including complex orography (Figs. 7b,d).

Hovmoeller diagrams of the snow precipitation over selected cross sections (red lines on Figs. 7a-d) confirm these day-night differences, offering more detail about the hourly variability of the snowfall (Figs. 7e-h). Over the western Cordillera, the first-day snowfall initiates during the afternoon and is maintained until early night, while the second-day snowfall is synchronized to the COL passage (Fig. 7e). Snowfall over southern Altiplano (Fig. 7f) starts in the north around 0000 UTC and then propagates south-eastwards in both days. The north-eastern Altiplano (Fig. 7g) shows snow precipitation after 0400 UTC (local midnight) every day. Finally, snowfall along the Andes-lowlands transition (Fig. 7h) is less homogeneous, but clusters mostly around the night.

### 3.3.1  Low-level circulation and moisture transport

Analysis of the 850 hPa equivalent potential temperature ($\theta_e$) gradient indicates the position of the cold front. We combine this information with the IVT and the relative humidity in order to assess the full column moisture circulation over the Altiplano and the cloud cover.

Figure 8a indicates a high initial southward IVT along the eastern Andean slopes. Once the cold surge enters the domain, moisture is advected towards the Altiplano along intra-andean valleys. At the same time the relative humidity increases following this circulation until reaching saturation (Fig. 8b). The cold front position stays quasi-stationary during the next 24 hours and the IVT is quite homogeneous all over the Altiplano and Pacific Ocean while still having some preference over the eastern Andean valleys. IVT appears to be unaffected by orography but additional analysis suggest that synoptic transport is dominant over low level fluxes (not shown). The atmospheric water vapour content is high at this moment everywhere for 12 more hours (Fig. 8b-c), consistent with satellite imagery (Fig. 1c). The end of the event is characterized by near-saturated mid-troposphere atmosphere propagating eastwards following the COL appearance (Fig. 8d). The IVT is still high, but increasingly directed south-eastwards.

500 hPa circulation analysis on the afternoon prior the start of the event (Fig. 9a) shows convergence over complex orography with an inside Altiplano prolongation from the west mountains. The convergence zone over the western Cordillera appears to propagate eastward over the Altiplano during the night (Fig. 9b-c). On the other hand, convergence over complex orography near Titicaca lake and La Paz city is still present. The first night was characterized by a relatively weak 500 hPa winds. Satellite image on August 24 at 0000 UTC (Fig. 9d) agrees with the convergence band shown in Fig. 9c that starts over to the lake surface. We compare this two variables since the simulated outgoing long-wave radiation (OLR) overlaps to the convergence zones (not shown)

The same analysis for the second night (25 August) shows an enhancement of 500 hPa winds (Fig. 9e-f). Therefore the convergence zones tend to be slightly less closely aligned with local topography and more affected by the synoptic circulation, although still showing a clear preference over complex orography. The satellite image on August 25 at 0000 UTC (Fig. 9g) also agrees with CTRL run and shows a more generalized cloud cover. The increase in wind speed culminates with the arrival of the COL over the Altiplano, accompanied by dry and strong winds (see next section).




Additional 3-day back-trajectory analyses over the domain D02, indicate that the north-western Amazon Basin serves as the main air income source. Selected places located along the Andes-Amazon transition received air masses from the south-western Amazon basin. There are also some trajectories that indicate the northward propagating cold surge east of the Andes, although this air is unlikely to be associated with major moisture transport as these air masses tend have very low moist static

energy (Hurley et al., 2015) (Fig 10a-b). More central Altiplano located regions shows a mix of sources coming generally from the north-west (Fig 10c-d). The first night (24 August at 0600 UTC, Fig. 10a,c) is characterized by a slow-moving circulation with air masses origination at low levels and following frontal and thermal up-slope motion preferably through the valleys, while upper-level air comes mostly from the upper atmosphere located over north-west Bolivia (Fig. 10a). Over the central Altiplano, arriving air masses also follows thermal circulation but coming mostly from the Pacific Ocean and from north-west

Altiplano (Fig. 10c). The circulation during the second night (25 August at 0600 UTC, Fig. 10a,c) shows a similar pattern as during the previous night, but generally stronger winds.

Concerning the vertical structure, we show cross-sections over the red solid lines plotted in Figures 7b (Fig. 11a-c) and 7c (Fig. 11d-e) corresponding to three days at 0600 UTC. The western region shows rapid surface cooling from 23 to 24 August (Fig. 11a-b), staying cold until 25 August (Fig. 11c). Vertical motion on the 24 August follows complex orography and cold

parcels (Fig. 11b) while on the 25 August it follows mountain gravity waves (Fig. 11c). The vertical motion over the eastern Altiplano follows mostly cold air regions propagating from the west, while the lake doesn't seem to exert a significant impact on atmospheric instability (Fig. 11d-f)

### 3.3.2   COL passage over the Andes

Cross sections over the Pacific Ocean show an atmospheric vertical structure consistent with synoptic COL development. WRF

simulations shows that the cyclone reached the surface over the Altiplano on 25 August at 1800 UTC, accompanied by strong and dry winds. This COL passage marked the end of the event and its characteristics are consistent with previous studies of such events in the region (Fuenzalida et al., 2005; Garreaud and Fuenzalida, 2007; Garreaud et al., 2003a). Thus, we decided to not to include these results here.

### 3.4   Sensitivity experiments

### 3.4.1   Orographic features

Reducing the topography of the Cordilleras surrounding the Altiplano (experiment RTA) resulted in a smaller amount of snowfall over the high mountains and a larger amount further to the south during the first day of snowfall. The second day event showed an opposite tendency and was accompanied by larger quantities of snowfall over the Altiplano and less snow to the south. The influence of the reduced topography on the resulting snow cover is small over the Altiplano, but it becomes

important at higher elevation, where the snow cover is reduced everywhere by around 20 %. A similar experiment, but with the elevation of the Altiplano also reduced (experiment RT80), confirms the importance of the mountains for influencing the local circulation over the Altiplano, but it also reaffirms the role of the Altiplano's height itself for determining the type of





precipitation. Not surprisingly, the RT80 experiment shows a general snowfall reduction all over the Altiplano and Cordilleras (Supplementary figure S1)

### 3.4.2 Titicaca lake influences

The removal and surface temperature modification of the lakes over the Altiplano leads to a change in the spatial distribution of

precipitation. We only show results for the 24. August snowfall during the night, because the most important thermal circulation influences occur at that time.

A colder lake (experiment LK-5, Fig 12a) restrict the airflow from the Amazon and weakens the nocturnal breeze towards the lake. The impact is to shift snowfall away from the lake along a NW-SE oriented band in the direction of more complex orography inside the Altiplano. At the same time, cross sections (Fig. 12b-c) show a limited influence of the lake for thermal

convection. The restriction of the Amazon airflow over the eastern edge of the Altiplano provokes an increase in snowfall over CP by around 10 mm (Fig 12a).

The NOLAKE experiment has a similar influence as the LK-5 experiment inside the Altiplano, albeit the air blocking from the Amazon, which now also affects the eastern Andean ridge and CP. The suppression of the nocturnal lake breeze influences the spatial distribution of snowfall by displacing the main snowfall band eastwards (Fig. 12d). Associated with this shift is the

decrease of snowfall over the Andes-Amazon transition by around 10 mm. Cross sections (Fig. 12e-f) shows similar features than in the cooler lake experiment.

## 4 Discussion

The snowfall event documented here occurred within the context of 2013 being a particularly cold winter in South America. A high number of snowfall days (11) were observed in southern Brazil (Mintegui et al., 2018) and several of the coldest air

intrusions in the twenty-first century were documented (Lanfredi and de Camargo, 2018).

The synoptic analysis suggests that a high atmospheric water vapour content over the north-western Amazon basin primed this event several days prior to its occurrence. The onset of the event was characterized by the establishment of a quasi-stationary Rossby wave over the Pacific Ocean and the appearance of a persistent 500 hPa high-pressure system over the Andes. At the same time, over the Amazon, there were remnants of a prior cold surge that suppressed the low-level jet. During

this phase, the southward water vapour transport was blocked by the frontal activity over the Amazon, resulting in the formation of an important atmospheric water vapour reservoir. This results are consistent with the findings of Sicart et al. (2015), who identified the Amazon region as humidity source for cloudinness over the eastern Cordillera.

The second phase included the southward water vapour transport and discharge phase; the IVT climatology analysis showed an intense water vapour transport towards and along the eastern Andean slope. This phase was initiated by the displacement

of the Andes' anticyclone synchronized with the Rossby wave displacement and thermal cyclogenesis over the Andes' eastern slope due to weak frontal activity. During this phase, we observed extreme amounts of IVT over the Andes that resulted in two days of heavy snowfall.





A short third phase was marked by the COL passage. While the Amazonian moisture transport was mainly controlled by the cold surge position and orographic influences. We believe the COL over the Pacific Ocean, besides the atmospheric destabilization effect, played a major role in the 500 hPa high-pressure displacement, facilitating moisture influx from the north-east towards the Altiplano until the final stage of the snowstorm and the later observed windstorm. Chronologically speaking, high-level PV fields propagated downwards from a trough axis and then completely segregated and intensified (forming the COL). The resulting impacts of this circulation at the surface included increased cloudiness and snowfall in the trough phase; and snowfall and strong winds during the COL phase.

The large-scale circulation appears to have been synchronized with the Rossby waves and the evolution of cold surges. While the relationship PV streamers/cold surges were previously documented over south-eastern Brazil (Sprenger et al., 2012), we find a similar pattern over the eastern slope of the Andes. We also note the importance of Rossby wave-breaking for COL formation. Our results confirm the snowfall mechanisms found by Vuille and Ammann (1997) (COL and cold fronts), and we do not find enough evidence for additional synoptic influences. We also note that, to the best of our knowledge, no major update for Vuille and Ammann (1997) study has been made.

Dynamically downscaled circulation during this event showed moisture flux towards the Altiplano occurring mostly along topographic depressions (valleys) connecting the Amazon with the Andes. Over the flat high-elevation plateau of the Altiplano, convection was generated by thermal circulation induced by orography and by lake-breeze circulation, combined with large-scale forcing depending on the time of the day. Daytime convection was accompanied by northerly and easterly winds originating over the SW Amazon and converging with westerly flow from the Pacific over the southern Altiplano. Nocturnal convection originated from convergence due to thermal circulation mixed with persistent north-westerly winds and air parcels originating from the Amazon.

Reducing the elevation of the highest peaks by half (experiment RTA) confirmed this behaviour by allowing the large-scale circulation to be more influential than the thermal circulation and by shifting the snowfall band towards the convergence zones. At the same time this configuration affected purely orographic snowfall, shifting them northwards following the large-scale wind sources. Moreover, reducing the full orography to 80 % of its original height (experiment RT80) results in the drastic reduction of snow cover overall. These experiments confirms the importance of elevated topography for snowfall distribution.

Experiment LK-5 suggest that the lake Titicaca influence for this event is twofold. (1) It shows the importance of the lake surface temperature for enhancing the diurnal moisture flow coming from the Amazon, in agreement with (Zängl and Egger, 2005), and (2) for shifting the convergence zones inside the Altiplano. Experiment NOLAKE confirm this with the nocturnal snowfall band shifted westwards (similar to LK-5) following the wind direction and allowing flows from the lowlands. We do not find sufficient evidence for thermal lake-effect snowfall.

## 5   Summary and conclusions

This paper investigated the principal characteristics of the synoptic and mesocale atmospheric circulation during a heavy snowfall episode over the central Andes in the austral winter of 2013. The storm developed during a cold surge episode and



it took place between 23 (night) and 25 (morning) August. Most snow fell during the night and it covered almost the entire Altiplano, albeit with higher intensity near the mountain sectors.

Synoptic analysis suggests three main elements that favoured the snowstorm formation, a) First, a strong prior cold surge generated a high-pressure system over the Amazon that directed moist air towards the Andes and an anticyclone over the Altiplano obstructing the upslope air flow. b) A second cold surge several days later triggered convection over the Amazon and it was accompanied by a pressure decrease over the Altiplano. The result is a combination of an overflow facilitating moist air transport towards the Altiplano c) An upper-level low pressure system to the west of the Andes providing a destabilizing mechanism and strong winds.

At a mesoscale level, numerical simulations reveal the important role of orography on both sides of the Altiplano and the influence of the Lake Titicaca on snowfall distribution. Our results suggest that: a) The high mountains provided an additional lifting mechanism to the cold front for convection and snowfall; at the same time, the valleys acted as a moist air passage towards the Altiplano. b) Concerning the Pacific ocean region, a high-level trough accompanied with a low level flow approached the west Andes; the mountains then blocked the airflow and the resultant gravity waves favoured snowfall over this region. c) The Lake Titicaca influences the circulation over the Altiplano through lake breeze circulation, but the nocturnal source of instability for convection over the Altiplano is not clear.

While our modelling experiments successfully reproduced the main features of this snowstorm. The spatial distribution in particular is succesfully reproduced numerically, and the model's configuration can be transferred to the SENAMHI for prediction purposes. However, it is still unknown to which extent our findings can be transferred to similar events. We offer an initial answer by studying an analog event in 2010, where the synoptic circulation provided the same main elements diagnosed in the earlier event. Nevertheless, given the low frequency of this type of events, our findings still need to be confirmed by further research both at synoptic and at mesoscale level.

*Competing interests.* The authors report that they do not have any conflict of interest.

*Acknowledgements.* This work was supported by the Swiss Federal Commission for Scholarships for Foreign Students in the form of a Swiss Government Excellence Scholarship (ESKAS No. 2015.0793). It has also received additional support from the project "Servicios CLIMáticos con énfasis en los ANdes en apoyo a las DEcisioneS" (CLIMANDES), no. 7F-08453.01.



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

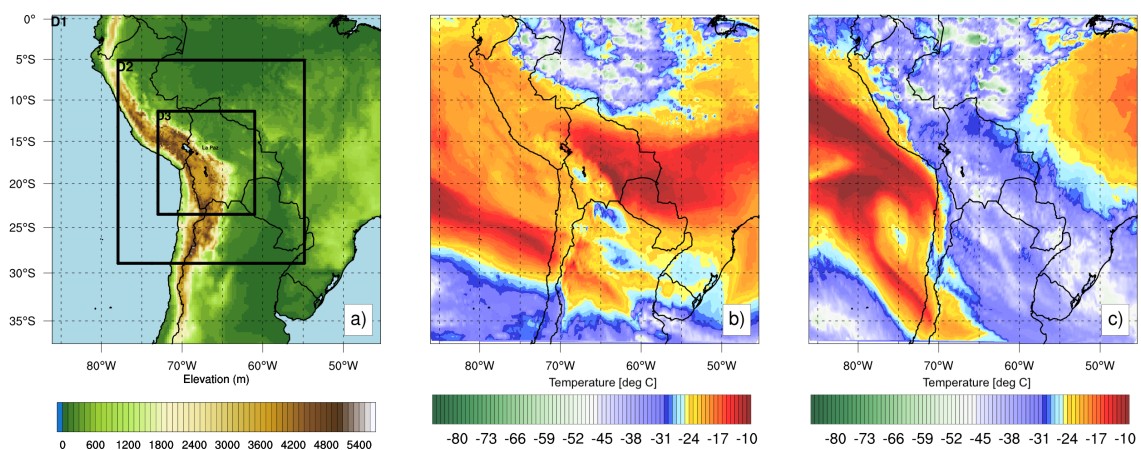

**Figure 1.** (a) Experiment spatial configuration with orography as seen by the model. (b)-(c) GOES-13 satellite images in the water vapour spectral band (6.48 µm) including brightness temperature [deg C] (coloured filled contours) at (b) 22 August 2013 at 0600 UTC and (c) 24 August 2013 at 0600 UTC

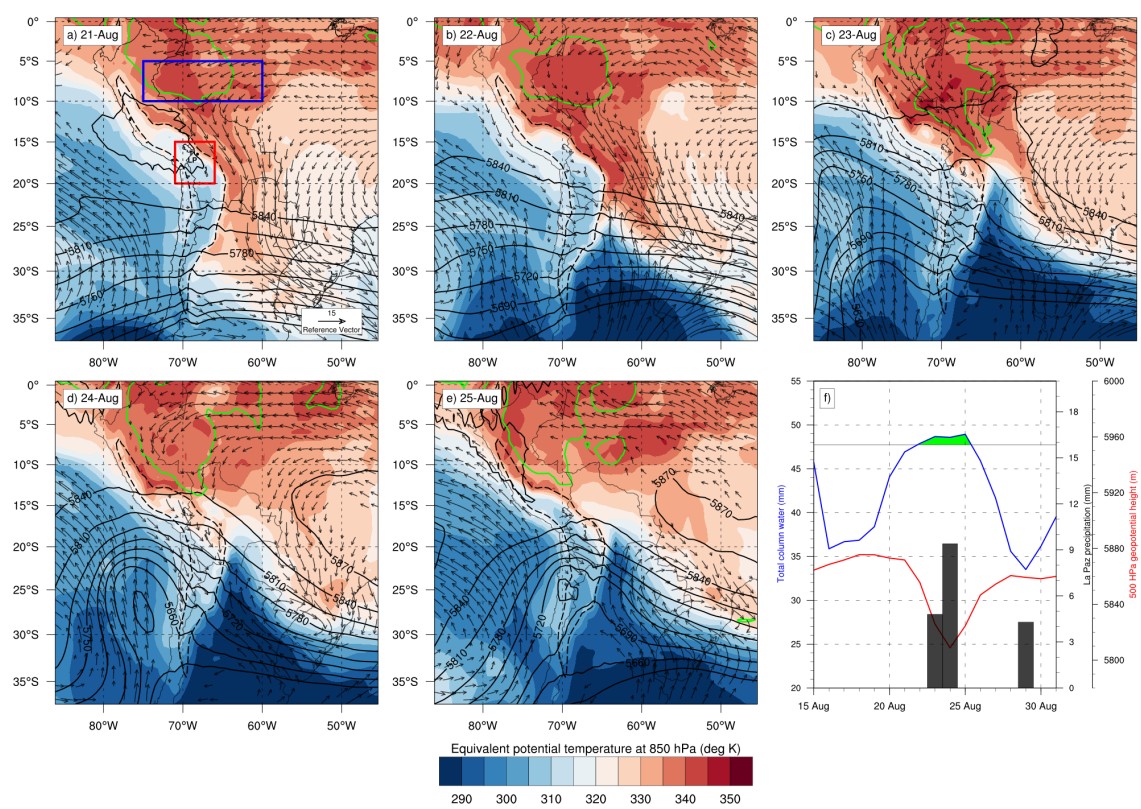

**Figure 2.** (a)-(e) 850 hPa $\theta_e$ [deg K] (colour shading) and wind motion in $ms^{-1}$ (arrows, wind with magnitude less than 3 $ms^{-1}$ is masked), and 500 hPa geopotential height [m] (black contours). Green contour shows the 85 percentile of PW over the blue box region. (f) Time series of the PW [mm] (blue line) over the blue box region with the 85 percentile (grey horizontal line) as reference, combined with the mean 500 hPa geopotential height [m] (red line) over the red box and the daily precipitation [mm] measured in La Paz (black bars)

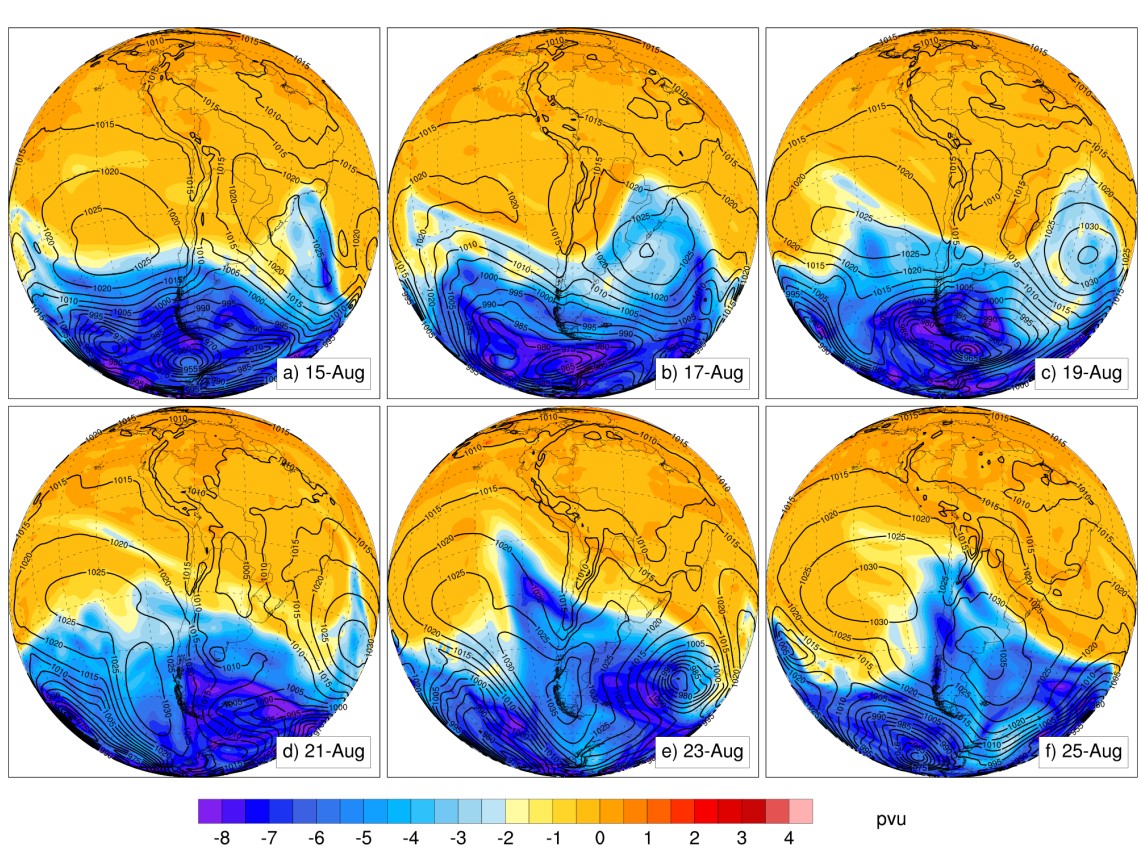

**Figure 3.** (a)-(f) Potential vorticity [PVU] at the isentropic level corresponding to 330 deg C (colour shading) and mean sea level pressure [hPa] (black contour). Plot is centred over 35S and 70W

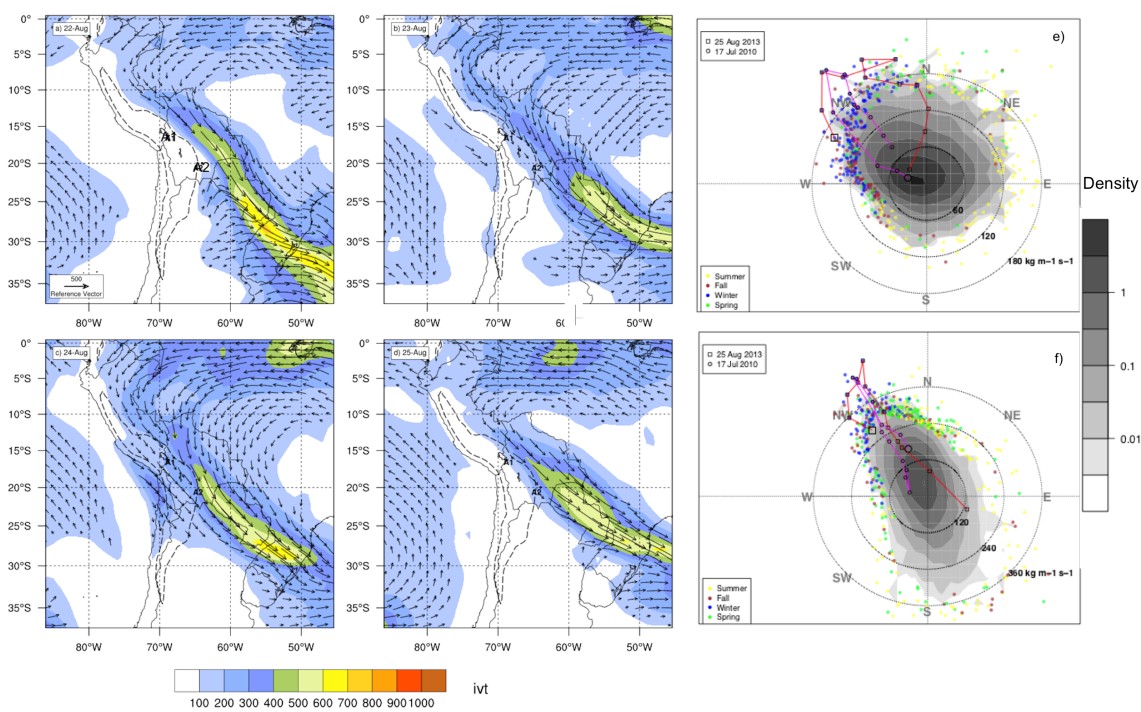

**Figure 4.** (a)-(d) IVT magnitude $[\mathrm{kg\,m^{-1}s^{-1}}]$ (colour shading) and direction (arrows). IVT arrows with magnitude less than $50\ \mathrm{kg\,m^{-1}s^{-1}}$ are masked. (e)-(f) IVT climatology at gridpoints A1 and A2, the diagram contains IVT magnitude (radius) and oncoming direction (angle) of every ERA-interim time step (6 hours) over 35 years. Count density is plotted on a logarithmic scale and coloured dots (colour indicate the season of occurrence) represents the values above the 99 percentile in each 5[deg] sector. The event chronologies are highlighted by the trajectories indicated by larger symbols.

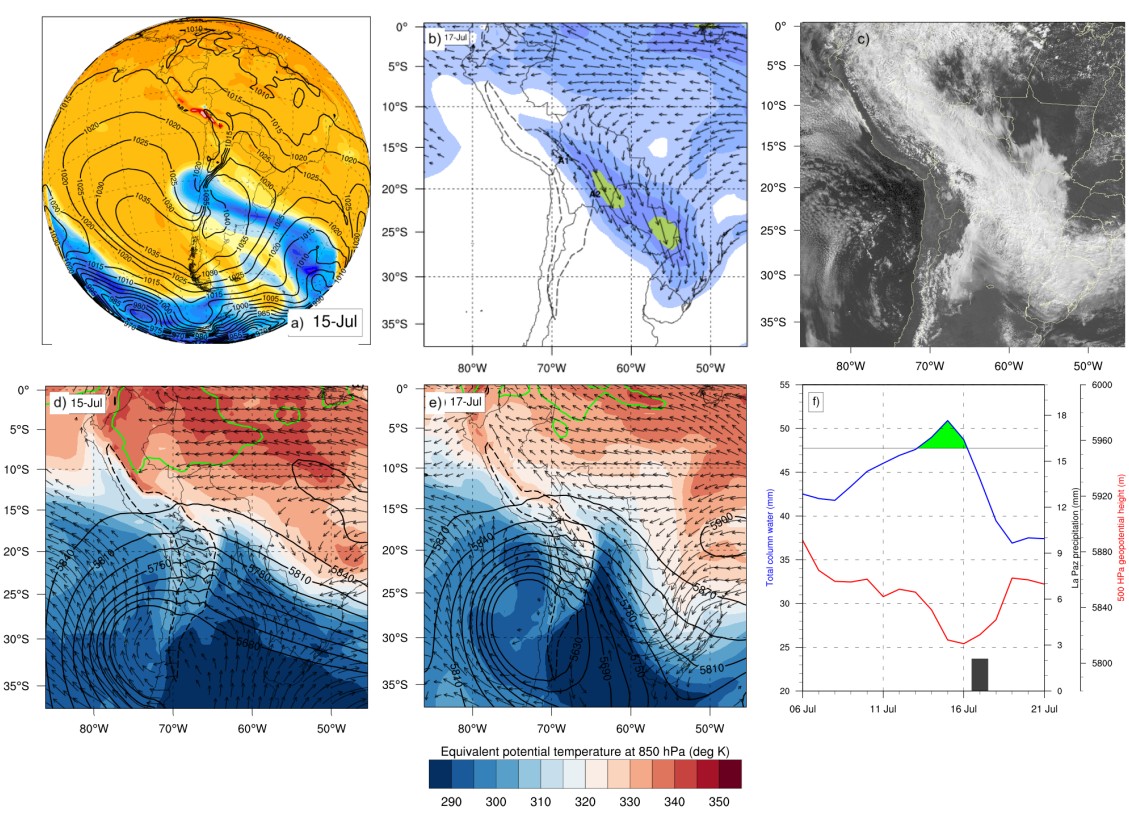

**Figure 5.** Analog Case study of 17 of July 2010. (a) As in figure 3, (b) as in figure 4a, (c) GOES-12 satellite image in the visible channel produced by the Center for Weather Forecast and Climate Studies of Brazil (CPTEC)–INPE (http://satelite.cptec.inpe.br/acervo/goes.formulario.logic, accessed 10 February 2019), (d)-(f) as in Figure 2d-f.

**Figure 6.** (a) 25 August MODIS snow cover [in percent] and (b) WRF simulated snow depth; both include the orography [m] (gray shading) as seen by the model. The location of the most affected areas (CP and TP) and the most populated cities (La Paz and Oruro) are indicated. c-d) Daily precipitation measurements on stations above 1500 meters from sea level for c) 23 to 24 August and d) 24 to 25 August

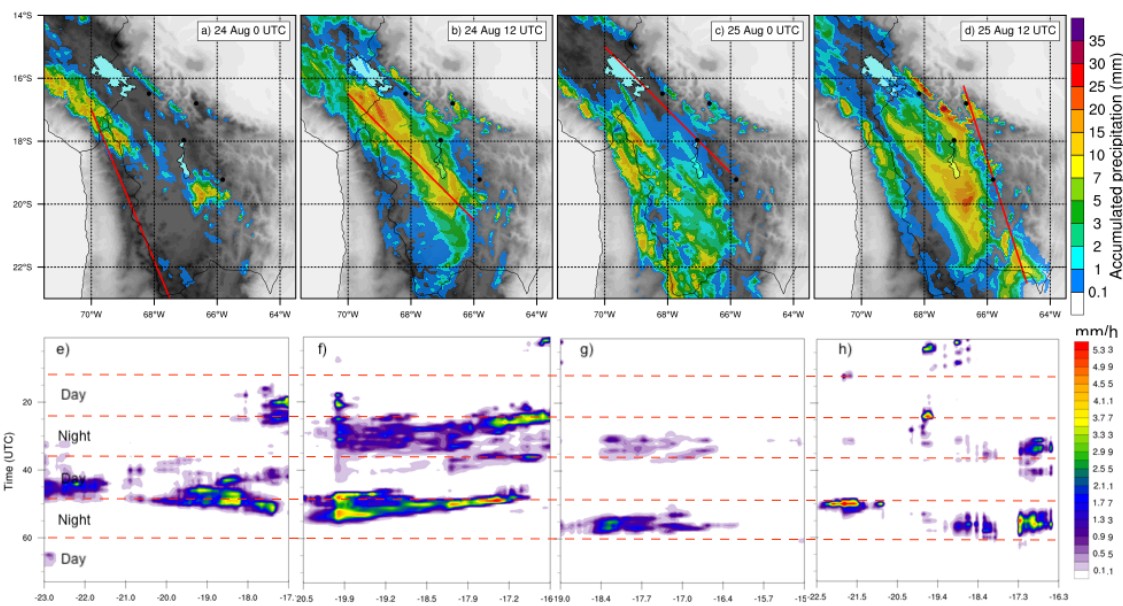

**Figure 7.** (a)-(d) 12 hour accumulated snowfall [mm] from CTRL run (colour shading). The time indicates the ending hour of accumulation and the solid red lines show the locations of the cross sections used for the plots below. (e)-(h) Snowfall Hovmoeller diagrams [$\mathrm{mmh}^{-1}$] over the corresponding cross section (colour shading). The dashed red lines are separated by 12 hours starting on 23 August 2013 at 0000 UTC

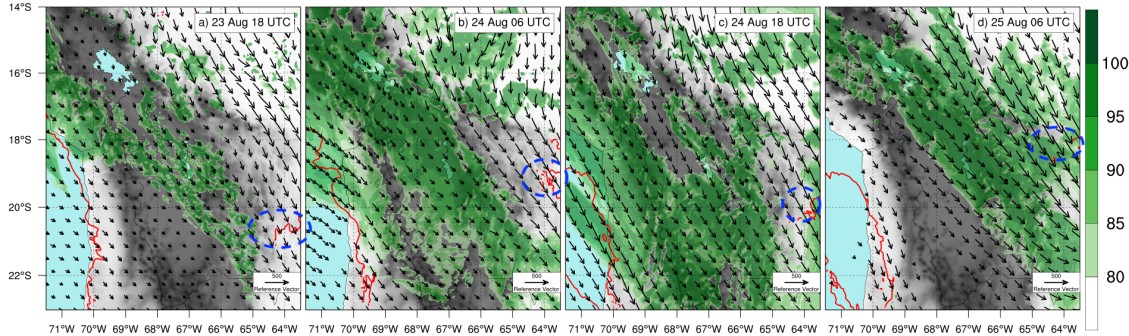

**Figure 8.** (a)-(d) CTRL run IVT [$\mathrm{kgm}^{-1}\mathrm{s}^{-1}$] magnitude and direction (arrows), 500 hPa relative humidity [percentage] (colour shading) and 850 hPa $\theta_{\mathrm{e}}$ 315 K isentrope (red contour) as front position proxy, highlighted by a blue dashed ellipse. Times are indicated in each plot.

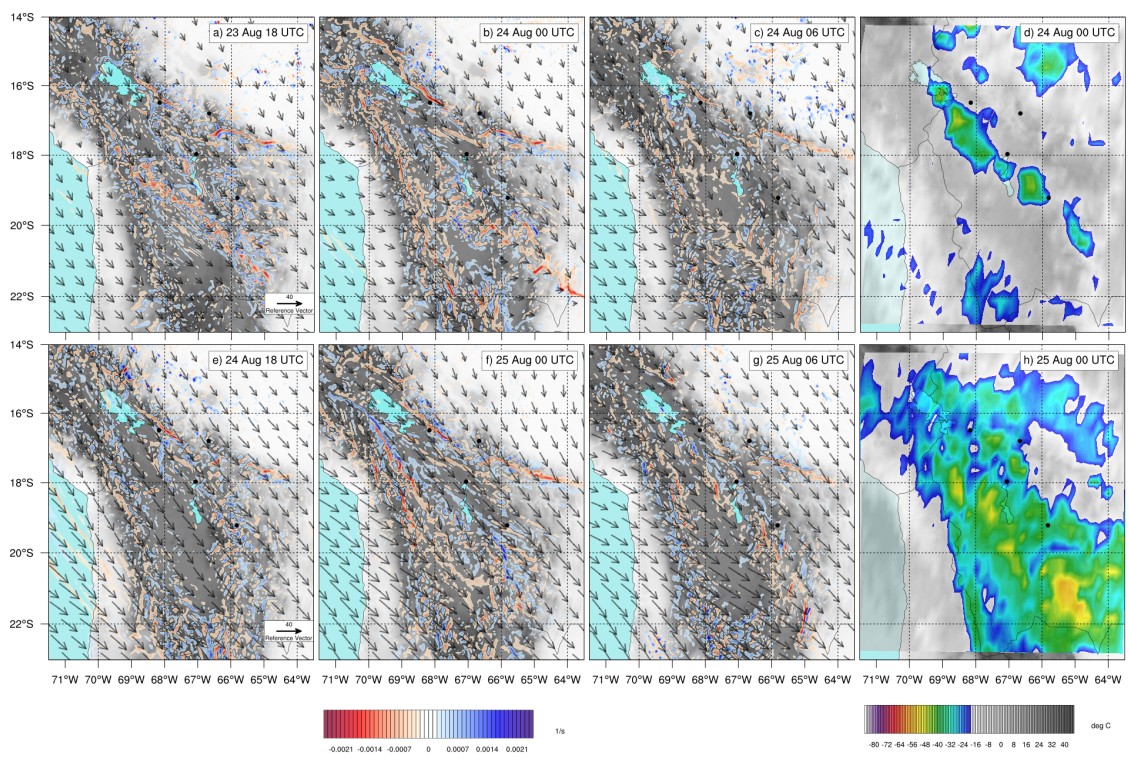

**Figure 9.** (a)-(c) and (e)-(g) 500 hPa divergence [s$^{-1}$] (colour shading) and wind speed [ms$^{-1}$] and direction (arrows); wind arrows with magnitude less than 5 ms$^{-1}$ are masked. (d) and (h) GOES-13 satellite images in the water vapour spectral band (6.48 μm) including brightness temperature [deg C] (colour shading). Times are indicated in each plot.




**Figure 10.** (a)-(d) 3 days backward trajectories with ending time on 24 August 2013 on 0600 UTC (left column) and August 25 at 0600 UTC (right column). Upper row shows the trajectories ending over La Paz and TP, while the bottom row trajectories end over Oruro and Uyuni salt flats. Colours indicates the elevation with respect to mean sea level. Each asterisk indicates a 3-hour timestep.

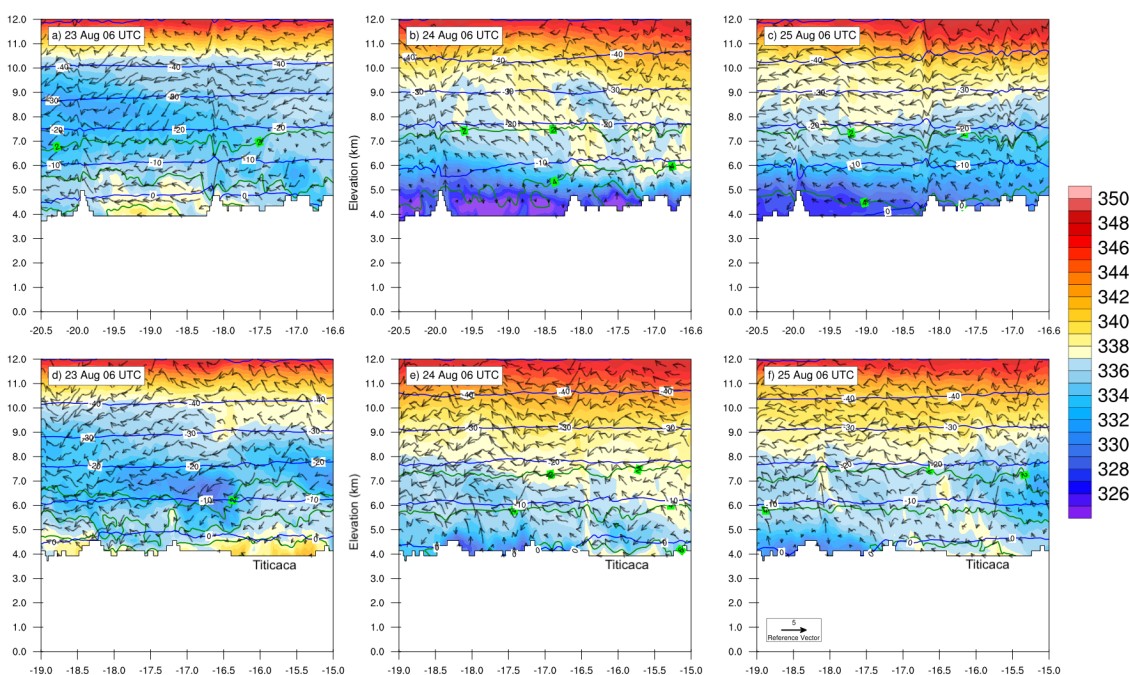

**Figure 11.** (a)-(f) Vertical cross section along transect marked by red solid lines shown in Figure 7b (upper row) and 7c (bottom row); including $\theta_e$ [deg K] (colour shading), water vapour mixing ratio [g kg$^{-1}$] (green contour), air temperature [deg C] (blue contour) and horizontal and vertical wind velocity [m s$^{-1}$] normalized by its standard deviation (curly vectors).

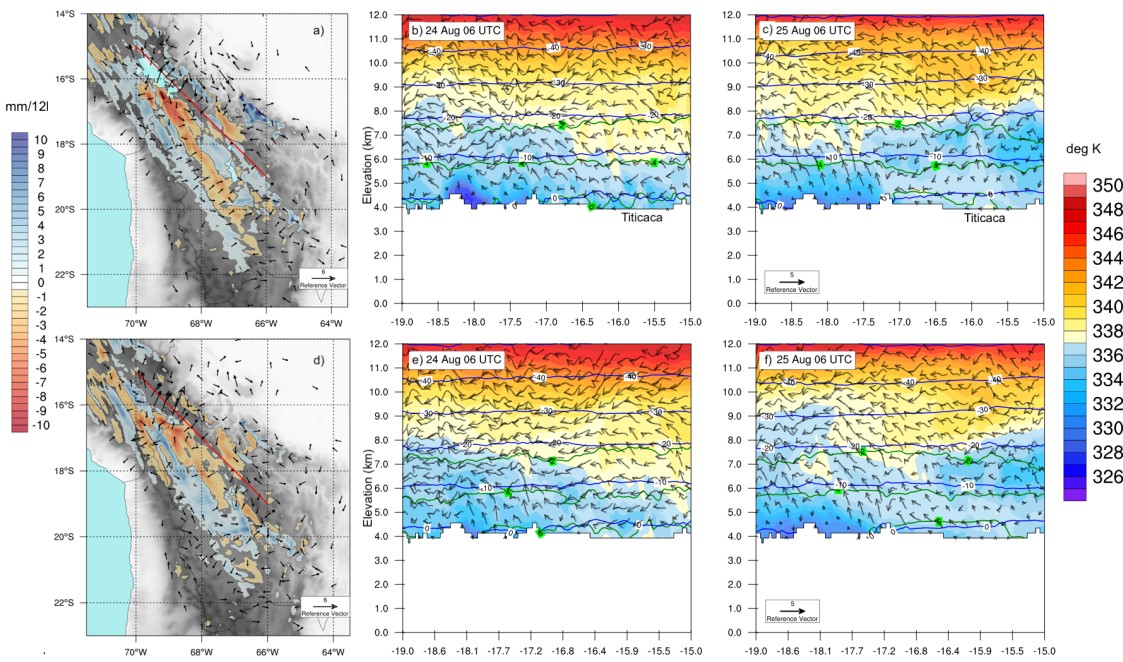

**Figure 12.** (a) and (d) 12 hours total snowfall and 10 meters wind mean differences between NOLAKE and CTRL experiment (a) and LK-5 and CTRL experiments (d). (b)-(c) and (e)-(f) as for Figure 11 over the corresponding cross sections indicated by the red solid line in Figs. 12a,d



**Table 1.** Main configuration used in WRF simulations

| Parametrization | Scheme |
|---|---|
| Cloud microphysics | New Thompson scheme |
| Planetary boundary layer | Yonsei University scheme |
| | with wind orographic correction |
| Land surface model | Noah-MP land surface model |
| | with Biosphere–Atmosphere Transfer Scheme |
| Radiation | Rapid radiative transfer model |
| | for longwave and shortwave |
| Cumulus parametrization | Kein-Fritsch scheme, |
| | parametrization off for D3 |

**Table 2.** Summary of WRF experiments

| Name | Goals | Description |
|---|---|---|
| CTRL | Real data configuration | Real orography, land use, and lake temperature |
| LK-5 | Sensitivity to lake temperature | Modification of lake surface temperature by -5 [deg C] |
| NOLAKE | Sensitivity to lake presence | Replacement of lake by the surrounding land use |
| RTA | Sensitivity to terrain reduction | Terrain above Altiplano level reduced by half |
| RT80 | Sensitivity to terrain reduction | Terrain reduced to 80 % of its original height |