# Peer review of "Synoptic and Mesoscale atmospheric features associated with an extreme Snowstorm over the Central Andes in August 2013"

_Natural Hazards and Earth System Sciences, 2019_

## Referee Comment (RC1) · Anonymous Referee #1 · 6 Nov 2019

General Comments

The authors provide a detailed study of a snowstorm case affecting the South America Altiplano during August 23, 2013. The analysis is supported by MODIS, TRMM, and GOES Imagery, ERA-Interim products, WRF simulations (3 km inner domain) and surface observations. With this information, the authors aim to describe the synoptic and mesoscale forcing that led to significant snow accumulation over the Altiplano.

Studies addressing the mechanisms that lead to extreme events are highly relevant in the current context of a changing climate. The numerical experiments performed by the authors are relevant to understand the influence that surface features (mountains and

lakes) have over the enhancement of snowstorms. However, the present study shows several flaws regarding meteorological concepts and methodology to support the hypotheses presented. Details of these flaws are presented in the specific comments along with some recommendations on how to alleviate them.

Specific Comments

1. Use of WV imagery for cloud cover interpretation (page 4, lines 7-8)

The authors use the water vapor band imagery from the GOES-13 satellite (central wavelength at 6.55 $\mu$m according to WMO ) to assess the cloud cover during the storm. This band retrieves the brightness temperature associated with the presence of water vapor in the mid to upper troposphere. A direct interpretation of cloud cover is not advised because an undersaturated atmosphere will not show clouds, despite the presence of water vapor. For a direct interpretation of cloud cover, authors should use the Infrared band (10.7 $\mu$m) or the Red band (0.65$\mu$m). Indeed, the latter is named as "visible" by the authors (section 2.1.2), which is inaccurate since a visible image (not band) is composed by three bands (blue, red, and green).

2. Use of equivalent potential temperature ($\theta$e) gradient at 850-hPa to identify frontal zones (page 4, lines 20-24)

This is a common technique to identify frontal zones and seems reasonable to use for the purposes of this study. Figures 2 and 5 show $\theta$e fields while the authors expect the reader to identify the frontal zone they argue exist in the study domain. I strongly suggest one of these two approaches:

a) Use a gradient threshold to identify them just as is done in the study referenced by the authors (Schemm et al, 2018).

b) Modify the color map employed for $\theta$e such that the temperature gradient concentration (indicative of a frontal zone) is clearly identified (see Figure 1 of Sprenger et al, 2012)

An analysis of $\theta$e at 850-hPa will indicate frontal position around the Altiplano and not over it. Please clarify if this level is used to support the idea of cold air advection over the Altiplano.

3. Use of potential vorticity (PV) analysis does not seem enterally justified in the text (page 4, lines 25-26)

I would think that many NHESS readers could be unfamiliar with the PV concept and its interpretation. I strongly suggest expand this two-line paragraph by adding some explanation about the PV analysis, especially about its interpretation (e.g. cold air intrusions from the stratosphere).

4. Use of integrated water vapor transport over the Altiplano (page 4, line 27-28)

The Altiplano surface is at $\sim$600-hPa but the methodology does not mention if this is considered when calculating IVT from surface over the Altiplano. Please state this clearly since it would affect the IVT results.

5. Cold fronts characterized by sea level pressure (page 6 line 20-21)

The statement "cold fronts position (characterized by high sea level pressure)" is wrong. If any, cold fronts are characterized by a strong horizontal temperature gradients and wind shift (Schultz 2005 MWR , Schemm et al 2018). As a result, it is hard to make the connection between the plume of -2 PVU at upper levels and the presence of a cold front near the surface, as the authors suggest.

6. IVT transport over the Altiplano (page 6, lines 25-30)

As mentioned in the comment n°4, a clarification of the IVT methodology is needed. The IVT analysis is suggesting transport from lower (Amazon) to higher (Altiplano) lands. Nevertheless, there is a significant altitude difference and therefore is hard to make a clear interpretation of the moisture transport in this context. As a counter example, IVT analysis have a straightforward interpretation over the ocean because is the same depth of atmosphere that contributes to the flux. Furthermore, it is hard

to distinguish which component is leading to strong moisture flux over the Altiplano: strong high-level winds? Or high-moisture content? The analyses provided in the study does not allow to a clear distinction between these components. I think that providing an analysis of the water vapor column anomaly could alleviate the IVT interpretation ambiguity.

7. WRF simulation timing assessment (page 7, line 21)

Although the authors claim that the WRF simulation was capable of reproducing the timing of the storm, no time series analysis is provided. I suggest either add a time series analysis or remove this statement.

8. Moisture advection along Andean valleys (page 8, lines 18-19)

The authors state "moisture is advected towards the Altiplano along intra-andean valleys". Although this seems a plausible hypothesis, no evidence is presented to support it. If Figure 8 is intended to be used for this purpose, the moisture flux associated with the valleys should be clearly portrayed, which is not the case in the current Figure 8. Even more, later the authors state that "IVT appears to be unaffected by orography" and that "synoptic transport is dominant", which seems contradictory with an intra- andean valley transport, where orography would be important. Besides, comment n°4 and 6 also hold for this statement.

9. Atmospheric water vapor content is high (page 8, lines 19-20)

Figure 8b-c shows relative humidity, which is not an indicator of absolute moisture or water vapor content. The atmosphere could have low moisture content and be saturated. Therefore, the statement "the atmospheric water vapor content is high" cannot be supported with Figure 8. In addition, GOES Imagery provides the moisture content at certain level and not over the full atmospheric column. There are other satellite products that provide this information . In addition, authors should clearly distinguish through the paper if they are referring to the moisture content of a specific

layer or in the atmospheric column.

10. Cloud cover in August 25 (page 8, line 33)

Same as previous comments, GOES water vapor imagery is not directly indicative of cloud cover.

11. Extreme amount IVT resulted in heavy snowfall (page 10, line 31)

From the analyses presented it is not clear that this is the case. Even if the IVT is indicating transport from the north towards the Altiplano, it is unclear if the absolute moisture is anomalously high, or if the Altiplano was anomalously cold.

12. Prediction purposes (page 12, lines 17-18)

If it is unknown the extent to which the current results can be transferred to similar events, what is the goal of transferring the model's configuration to the SENAMHI for prediction?

Technical corrections

Lines 14-15 on page 3 are not necessary.

Page 3 line 20 should read: "dataset produced by".

URL on page 3 line 22 should be moved to a foot note.

Page 3 line 26, the reference of SENAMHI should follow same pattern as the other references: SENAMHI (year).

Page 4 lines 7-8 should belong to the previous paragraph.

Page 4 line 16, please rephrase "Additionally, . . ." to clarify its meaning.

Page 5 lines 8-9, move the two-line paragraph to section 2.2.2 since this correspond to a WRF analysis.

Page 5 lines 16-19, move the extra content to Table 1. In other words, these lines are

not necessary since you have Table 1. This table only needs to be updated with the extra information contained in the abovementioned lines.

Page 5 line 28, replace "lake it by" for "lake for".

Page 6 line 20, use K instead of deg C.

Page 10 line 5, remove period in "24. August".

Page 11 lines 1-2, the clause "While…" is incomplete.

Page 13 line 28, first author is duplicated.

Page 13 lines 32-33, reference is duplicated.

Page 13 line 35, add title of the reference.

Page 15 line 7, URL is duplicated

Page 15 lines 28-30, authors and URL are duplicated

In all corresponding figures need to mask the Altiplano region for analysis at 850-hPa, as in Sprenger et al. (2012).

Use brighter colors in Figures 2 and 5 to clearly identify contours, arrows, and annotations.

Use larger fonts in all figures.

Provide larger figures (e.g. page width) so it is easier to distinguish thin lines (e.g country limits).

Need to add the meaning of the dashed line in Figure 2 and 4. I presume is the contour of Altiplano at certain altitude.

Indicate the data source in each figure. For example, it is unclear from the caption in Figure 2 if these are GFS or ERA-Interim data.

[Figure]

Figure 4 need larger annotations for A1 and A2. Also, panels e) and f) need an annotation to indicate to which site correspond each.

Figure 5 need date and time in panel c). In addition, need the timestamp in all analyses (review previous and following figures) unless they are daily composites, which should be stated in the methodology and figure's caption.

Figure 11, why using curly vectors? If no physical explanation is provided then authors should use regular vectors.

---

## Referee Comment (RC2) · Anonymous Referee #2 · 12 Nov 2019

General comments:

This paper presents an analysis of a snowstorm event that occurred over the central Andes. The aims are to characterise the synoptic evolution and mesoscale processes leading to the event as well as to determine the specific roles of the orographic features and lake Titicaca. ERA-Interim reanalysis data and observations are used for the synoptic analysis whereas WRF model simulations including sensitivity studies are used to analyse the mesoscale features and the roles of the orography and lake. The authors have synthesised an impressive collection of reanalysis data, model output and observational data. This synthesis yields a two-stage mechanism for the synoptic evolution

and information on the factors controlling the mesoscale details of the snowfall regions. The topic appears suitable for NHESS. However, the paper requires fairly substantial polishing before it will be ready for publication. The text is perfectly understandable, but has many small English language errors (some, but not all of which, I've indicated below). Some of the figures also need correcting/improving. In several places I struggled to relate the figures to the associated interpretation in the text. Finally, the novelty of the work, its implications, and its relationship to other studies needs to be made clearer. I was left wondering how much the findings could be generalised to other mountain ranges and cases or were unique to this specific event.

Major specific comments:

**p6, L7:** Here you infer cloud formation and cover from plots of water vapour imagery brightness temperature. I think you need to translate for the reader - i.e. cold brightness temperatures imply high cloud tops. I am also struggling to relate your text to the figures: e.g., you say that on the 22nd there is cloud formation over northern Bolivia but the brightness temperature is relatively warm there. Why do you not instead show the visible image at a suitable time as you do for the analogue case in fig. 5c? Also, why do you not use the same colour scale as in fig. 9d and h which is also GOES-13 data?

**p6, L11:** Similarly, here you discuss the total water column displacement and a quasi-stationary cold front relating to Fig 2. But, what you show in fig. 2 is $\theta_e$, winds and precipitable water. Please help the reader by relating the features you discuss in the text to the fields plotted and also avoid changing terminology e.g. between total column water to precipitable water.

**p6, L20:** Here you say that the position of the cold front is characterised by high sea level pressure – yes I can see that there are mean sea level pressure contours that align with the Andes but I'm afraid that I can't work out where the cold fronts

are from them and as the projection of this plot (and days shown) differ from that used in fig. 2 it's hard to work out exactly where the cold fronts are. Perhaps you could mark them on the plots?

**p6, L21:** What exactly do you mean by "in phase with Rossby wave trains"? The PV structure on the 15th looks very different to that on the 23rd but your text implies that they are similar. Also it is the -2 PVU contour which marks the wave train rather than air with PV less than -2PVU (which instead indicates stratospheric air). Also change "portrayed by PVU lower than -2 units" to "portrayed by the -2 PVU surface".

**Section 3.1.1:** You show several plots of model fields in this section but don't state in the captions where the data is from; presumably it is from ERA-Interim reanalysis. Please be clearer about which plots use ERA-I and which your WRF model runs.

**Conclusions:** At the end of the discussion and conclusions sections I'm still unclear as to the novel results that have come out of your study. Yes, you have addressed the goals laid out in your introduction (to study this event at both the synoptic and mesoscales and to assess the importance of the orography height and lake). However, it's not clear the extent to which this synoptic evolution and the more local impacts of lakes and the orographic height were already known. Please can you more clearly state the novel contributions of your study.

Minor specific comments:

**Abstract:** The abstract describes the work completed and findings well. However, I'm left wondering about the implications of the work - could the authors add a final sentence that answers the "so what" question?

**General:** The text is often written as lots of short paragraphs rather than being grouped together into longer paragraphs on a specific top (see especially the introduction

which consists of 9 paragraphs, mostly 2 or 3 sentences long each). I appreciate that this is perhaps the writing style of the authors, but it comes across as note like rather than final text. The text would be easier to read if written as fewer, longer paragraphs.

**p5, L13:** Change "resolution" to "grid spacing" — the scale of features that a model can resolve is several times (often taken as at least 6 times) the grid spacing.

**p28, table 1:** This table doesn't seem to add anything to the details given in the text in section 2.2.2 and could be omitted.

**Figure 1:** It would help readers for the countries and regions mentioned in the text to be labelled on one of the maps, perhaps best on Fig. 1a.

**Figure 2:** Is the dashed line indicating where the weak winds are masked? This line encompasses the Andes so is it actually where the winds are weak or instead where the 850 hPa surface is below the ground surface? Also note that the acronym PW isn't defined anywhere.

**Fig 4e and f:** The symbols and text on these panels are very small and I had to zoom in on the pdf to be able to work out which set of joined symbols corresponded to which date - can you make this clearer please? Also, presumably the larger square and circle refer to some start date with the subsequent symbols 6 hours apart until an end date. Please explain this more clearly in the caption.

**Figure 6:** The labelling on these figure panels is useful but it's quite hard to read black text on a grey background (and the text is also quite small in places) Also, panels a and d have "July" written on them whereas the caption says that they are for August.

**Section 3.2.1.:** Here you discuss results from your control WRF run but this discussion would probably be better included in the following section (3.3) where you discuss

other results from your WRF control run.

**Figure 7 e-h:** The y-axis label is wrong as the hours are not UTC (the numbers exceed 24). However, for comparison with the text it would be more useful to have the numbers in UTC. Also, I'm confused by the labelling of night and day on these panels. The caption says that the first red dashed line is 0 UTC on the 23rd and so the space between the 1st and 2nd red dashed line covers 0-12 UTC on the 23rd which according to the text (p7, L31) should be night time, but it is labelled as day. Presumably instead the top of the y-axis is at 0 UTC?

**Figure 8:** In the text (start of section 3.3.1) it says that the 850 hPa $\theta_e$ gradient is used to indicate the front location which is fine. However, the plot instead shows a single isentrope of $\theta_e$. A small section of this isentrope is then "highlighted by a blue dashed ellipse" to indicate the front position. I'm afraid that I don't understand how the position of the front can be determined from this small contour section. Nor do I understand how a small highlighted region indicates the location of a large-scale, generally linear feature such as a front.

**Figure 9:** The font size used for the numbers on the colour bars in this figure particularly are tiny. Please can the text be enlarged. I encourage the authors to look again at the font size used in all of their plots as there are other places too where the font size (particularly on labels) is too small to be easily read in a printed version of the paper. Also, what is the blue shading (particularly over the ocean) and underlying grey shading on panels a-c and e-g? Do they indicate water and topography, respectively? Please label Titicaca lake and La Paz city on one of the plots as these locations are mentioned in the corresponding text.

**p8, L25:** Here it says that the convergence zone over the western Cordillera appears to propagate eastwards during the night. From Fig. 9 though it appears more that the convergence simply weakens during the night.

**p8, last two paragraphs:** The description of "night" in these paragraphs is somewhat inconsistent with the earlier definition of local day as being 12-0 UTC and night 0-12 UTC. The corresponding figure panels from which the nighttime evolution is described are at 18, 0, and 6 UTC. Would it be better to show them at 0, 6, and 12 UTC?

**Figure 11:** Please label the axes.

**p6, L16:** Here it says that "the lake doesn't seem to exert a significant impact on atmospheric instability". However, Fig, 11d-f shows that $\theta_e$ decreases in height immediately above the lake, particularly on the 23 Aug. Hence this air is potentially unstable.

**p10, L11:** Please mark CP also on one of the maps in figure 12.

**p11, L5:** The phrase "high-level PV fields propagated downwards from a trough axis" doesn't make sense. I think you mean that air with high PV values propagated downwards, but I'm also not sure that "propagating" is an appropriate term here.

**p11, L9:** The phrase "The relationship PV streamers/cold surges were . . . " is missing a word. Do you mean "The relationship between PV streamers and cold surges were . . . "?

Technical errors:

Note that not all of the small English language errors are included here.

**Abstract and elsewhere:** Usually adjectives should be hyphenated before a noun e.g. "large-scale analyses", "low-level blocking". This seems to have been done a bit randomly in the abstract at least (i.e. sometimes the adjectives are hyphenated and other times they are not).

**Abstract and p3, L9:** Change "2013's" to "2013".

**p2, first line:** change semicolon to a comma.

**p2, L13:** change "circulation" to "circulations".

**p2, L18:** remove comma after gap.

**p2, L29:** change "emergency state" to "state of emergency".

**p3, L8:** change "introdude" to "introduce".

**p3, L10 and 11:** change "contain" to "contains", "summarize" to "summarizes", and "include" to "includes".

**p3, L15:** Change to "For this study about heavy snowfall over complex orography we used the following datasets".

**p3, L17:** Add space after "Andes".

**p3, L29:** Remove brackets around "SENAMHI".

**p4, L4:** Change "conditions" to "condition".

**p4, L8:** Change to "for a spatial assessment of cloud cover".

**p4, L10:** Change "asses" to "assess".

**p4, L12:** Change to "from the surface".

**p4, L:23** Add space after "processes".

**p5, L15:** Change "runs" to "run".

**p5, L22:** Spelling "lengths".

**p5, L28:** Remove "it".

**p5, L29:** Replace "are described" by "is".

**p6, L15:** Change "correspond" to "corresponds".

**p6, L26:** Change "by" to "on".

**p7, L26:** Change to "we remind the reader that".

**p8, L28:** Change "this" to "these".

**p9, L5:** Change "shows" to "show".

**p9, L7:** Change "origination" to "originating".

**p9, L9:** Change "follows" to "follow.

**p9, L20:** Change "shows" to "show".

**p10, L5:** Remove full stop after "24".

**p10, L7:** Change "restrict" to "restricts".

**p10, L15:** Change "shows" to "show".

**p10, L16:** Change "than" to "to".

**p11, L13:** Add "the" after "for".

**p11, L23:** Change "them" to "it".

**p11, L26:** Change "suggest" to "suggests".

**p11, L28:** Change "confirm" to "confirms".

**p11, L29:** Change "shifted" to "shifting".

**p12, L3:** The punctuation in this paragraph and the following is confused. For example a list following a colon needs to be separated by commas or semicolons.
* * *

---

## Referee Comment (RC3) · Anonymous Referee #3 · 26 Nov 2019

General Comments This manuscript provides a detailed overview of the synoptic and mesoscale patterns associated with a high impact snowstorm over the Andes of southern Peru and Bolivia in August 2013. The manuscript is very well organized and generally well written, although many paragraphs are quite short and could be expanded/combined. The data, methods, and analyses are all appropriate and conclusions are consistent with the results. The discussion section, however, could be strengthened considerably, as there are only a handful of other papers referenced, and comparison of results with other published literature could be beneficial. Many of the figure panels are very small and difficult to read/interpret and some enlargement could be helpful.

Specific Comments p. 2, line 2: A number of recent studies (e.g., Romatchske and Houze 2013, Mohr et al. 2014, Endries et al. 2018) have demonstrated that much of the precipitation in the central Andes of Peru and Bolivia is stratiform and not exclusively convective. p. 2, line 23: How is the spatial distribution of snowfall clear? p. 4, line 16: Incomplete sentence? p. 7, lines 15-19 and Fig. 6: The grey shading is somewhat confusing as there is a class of >100% and >1.0 m corresponding to grey in Figs. 6a and 6b. Suggest removing this grey class from the legend. The colors for the graduated circles are also a bit hard to interpret but this is partly ameliorated by the graduate size. Adding an inset map for the La Paz vicinity could help? It could be placed in the southern Altiplano/northern Chile? p. 9, lines 1-10 and Fig. 10: It is not clear what the ending heights (pressure or amsl) for the different panels. Please clarify. p. 11, lines 17-20: What is the rationale in support of there being daytime and nighttime convection during this event? If no strong evidence then suggest changing "convection" to "precipitation" or "snowfall." p. 12, lines 14-15: Again, what evidence is there to support the assertion that there was nighttime convection? Per first comment, recent work has demonstrated that nighttime precipitation across the region is mainly stratiform. Are there any manual or acoustic snowfall observations from Chacaltaya or Zongo in Bolivia?

Technical Comments p. 3, line 20: Consider replacing "trustful" with "reliable"? p. 4, line 4: s not needed in "conditions" p. 8, line 18: "analysis suggest" should be changed to "analysis suggests" p. 9, line 16: Avoid use of apostrophes (doesn't) p. 10, line 5: Period not needed after 24 p. 10, line 7: Change "A colder lake restrict" to "A colder lake restricts" p. 11, line 9: Missing "of" between "relationship" and "PV"? p. 11, line 13: Missing "the" between "for" and "Vuille"? p. 12, line 16: Incomplete sentence

---

## Author Comment (AC1) · 17 Jan 2020

General Comments

The authors provide a detailed study of a snowstorm case affecting the South America Altiplano during August 23, 2013. The analysis is supported by MODIS, TRMM, and GOES Imagery, ERA-Interim products, WRF simulations (3 km inner domain) and surface observations. With this information, the authors aim to describe the synoptic and mesoscale forcing that led to significant snow accumulation over the Altiplano. Studies addressing the mechanisms that lead to extreme events are highly relevant in the current context of a changing climate. The numerical experiments performed by the authors are relevant to understand the influence that surface features (mountains and lakes) have over the enhancement of snowstorms. However, the present study shows several flaws regarding meteorological concepts and methodology to support the hypotheses presented. Details of these flaws are presented in the specific comments along with some recommendations on how to alleviate them.

We thank the reviewer for the time invested in reading this manuscript. We will be happy to take into account most of the comments and recommendations herein in order to improve the clarity of the evidence that support our hypotheses.

Specific Comments
1. Use of WV imagery for cloud cover interpretation (page 4, lines 7-8)
The authors use the water vapor band imagery from the GOES-13 satellite (central wavelength at 6.55 μm according to WMO ) to assess the cloud cover during the storm. This band retrieves the brightness temperature associated with the presence of water vapor in the mid to upper troposphere. A direct interpretation of cloud cover is not advised because an undersaturated atmosphere will not show clouds, despite the presence of water vapor. For a direct interpretation of cloud cover, authors should use the Infrared band (10.7 μm) or the Red band (0.65μm). Indeed, the latter is named as "visible" by the authors (section 2.1.2), which is inaccurate since a visible image (not band) is composed by three bands (blue, red, and green).

Thank you for this comment. Since we analyze the water vapor content in many parts of the manuscript, we chose to use this band imagery. The reviewer is correct that for a direct interpretation of cloudiness, the Infrared band (10.7 $\mu$m) or the Red band (0.63 $\mu$m) should be used. We are here in front of two options: to use the Infrared band since it can detect cloudiness during night or to rewrite the text in base of water content instead of cloud cover. The revised manuscript will address this issue.

Regarding the use of wording of the 0.63 $\mu$m band, we based our nomenclature ("visible") in a document produced by the CIMSS from the university of Wisconsin. We agree that visible images are composed by three bands (blue, red and green), and we will clarify this in the revised manuscript.
Please read the document here:
http://cimss.ssec.wisc.edu/goes/webapps/bandapp/GOES_Imager_Spectral_Bands_overview.pdf

2. Use of equivalent potential temperature (θe) gradient at 850-hPa to identify frontal zones (page 4, lines 20-24) This is a common technique to identify frontal zones and seems reasonable to use for the purposes of this study. Figures 2 and 5 show θe fields while the authors expect the reader to identify the frontal zone they argue exist in the study domain. I strongly suggest one of these two approaches:
a) Use a gradient threshold to identify them just as is done in the study referenced by the authors (Schemm et al, 2018).
b) Modify the color map employed for θe such that the temperature gradient concentration (indicative of a frontal zone) is clearly identified (see Figure 1 of Sprenger et al, 2012)
An analysis of θe at 850-hPa will indicate frontal position around the Altiplano and not over it. Please clarify if this level is used to support the idea of cold air advection over the Altiplano.

Thank you for the suggestions. We will gladly use the second proposed approach since we aim to relate the position of the front to moisture transport towards the Altiplano and not front identification. A paragraph also will be added concerning the use of the 850-hPa level and the implications of the front position for moisture transport and lifting from the lowlands to the Andes.

3. Use of potential vorticity (PV) analysis does not seem enterally justified in the text (page 4, lines 25-26) I would think that many NHESS readers could be unfamiliar with the PV concept and its interpretation. I strongly suggest expand this two-line paragraph by adding some explanation about the PV analysis, especially about its interpretation (e.g. cold air intrusions from the stratosphere).

This point is important for explaining the strong winds that were observed after the snowfall event. The reviewer raises a good point by saying that the PV concept and its interpretation can be unfamiliar to some readers. We will add a sentence about this.

4. Use of integrated water vapor transport over the Altiplano (page 4, line 27-28) The Altiplano surface is at ~600-hPa but the methodology does not mention if this is considered when calculating IVT from surface over the Altiplano. Please state this clearly since it would affect the IVT results.

Equations (1) and (2) states that the IVT is calculated from the surface. The reviewer reminds us a good point by saying that the Altiplano surface is very high (~600 hPa) and it is worth to state this clearly. For this purpose we will rewrite this part.

5. Cold fronts characterized by sea level pressure (page 6 line 20-21)
The statement "cold fronts position (characterized by high sea level pressure)" is wrong. If any, cold fronts are characterized by a strong horizontal temperature gradients and wind shift (Schultz 2005 MWR , Schemm et al 2018). As a result, it is hard to make the connection between the plume of -2 PVU at upper levels and the presence of a cold front near the surface, as the authors suggest.

The reviewer makes a good observation by pointing the inaccuracy of this statement. While we were trying to make a point about the relationship cold front/-2 PVU plume, we may have chosen the incorrect variable. We will update Fig.3 and rewrite the text accordingly.

6. IVT transport over the Altiplano (page 6, lines 25-30) As mentioned in the comment n○4, a clarification of the IVT methodology is needed. The IVT analysis is suggesting transport from lower (Amazon) to higher (Altiplano) lands. Nevertheless, there is a significant altitude difference and therefore is hard to make a clear interpretation of the moisture transport in this context. As a counter example, IVT analysis have a straightforward interpretation over the ocean because is the same depth of atmosphere that contributes to the flux. Furthermore, it is hard to distinguish which component is leading to strong moisture flux over the Altiplano: strong high-level winds? Or high-moisture content? The analyses provided in the study does not allow to a clear distinction between these components. I think that providing an analysis of the water vapor column anomaly could alleviate the IVT interpretation ambiguity.

Thank you for this suggestion. We agree that IVT analysis can be tricky over a place with such a big change in altitude. It is true that there is not a clear distinction between the components' degree of participation (winds or moisture) and we will explore your suggestion. For this purpose, the revised manuscript will update Fig.8 and modify the text accordingly to make it also consistent with comment 4.

7. WRF simulation timing assessment (page 7, line 21) Although the authors claim that the WRF simulation was capable of reproducing the timing of the storm, no time series analysis is provided. I suggest either add a time series analysis or remove this statement.

Thank you for pointing out this error. We were aiming to compare the timing of the satellite's snow cover observation and the WRF simulation. We will rephrase this sentence to better express our interpretation.

8. Moisture advection along Andean valleys (page 8, lines 18-19) The authors state "moisture is advected towards the Altiplano along intra-andean valleys". Although this seems a plausible hypothesis, no evidence is presented to support it. If Figure 8 is intended to be used for this purpose, the moisture flux associated with the valleys should be clearly portrayed, which is not the case in the current Figure 8. Even more, later the authors state that "IVT appears to be

unaffected by orography" and that "synoptic transport is dominant", which seems contradictory with an intra- andean valley transport, where orography would be important. Besides, comment n○4 and 6 also hold for this statement.

We understand the reviewers' concern with our IVT analysis. We agree that a better analysis have to be made in order to offer more solid evidence to support our hypotheses. We will review our statements all over the manuscript concerning the IVT in accordance to comments 4 and 6.

9. Atmospheric water vapor content is high (page 8, lines 19-20)
Figure 8b-c shows relative humidity, which is not an indicator of absolute moisture or water vapor content. The atmosphere could have low moisture content and be saturated. Therefore, the statement "the atmospheric water vapor content is high" cannot be supported with Figure 8. In addition, GOES Imagery provides the moisture content at certain level and not over the full atmospheric column. There are other satellite products that provide this information . In addition, authors should clearly distinguish through the paper if they are referring to the moisture content of a specific layer or in the atmospheric column.

Thank you for this comment. The reviewer is correct that a better distinguish about moisture, water vapor and cloudiness (and their respective atmospheric level) has to be made. The reviewed manuscript will take better care of it. Concerning Fig.8, it will be modified as stated in our reply to comment 6 and the text will be updated accordingly.

10. Cloud cover in August 25 (page 8, line 33) Same as previous comments, GOES water vapor imagery is not directly indicative of cloud cover.

Same reply as for comment 9.

11. Extreme amount IVT resulted in heavy snowfall (page 10, line 31)
From the analyses presented it is not clear that this is the case. Even if the IVT is indicating transport from the north towards the Altiplano, it is unclear if the absolute moisture is anomalously high, or if the Altiplano was anomalously cold.

The reviewer makes a good observation. We will add a sentence about the atmospheric temperature.

12. Prediction purposes (page 12, lines 17-18) If it is unknown the extent to which the current results can be transferred to similar events, what is the goal of transferring the model's configuration to the SENAMHI for prediction?

This is a fair point. While the analog case of 2010 is not enough evidence for generalizing our results, we still believe that the Senamhi can benefit from this study. We will rewrite this sentence to better justify this conclusion.

Technical corrections

Thank you for pointing out the technical errors. We will take into account those that stay in the manuscript after addressing the specific comments (including the updated figures)

Lines 14-15 on page 3 are not necessary.
Thank you
Page 3 line 20 should read: "dataset produced by".
Thank you
URL on page 3 line 22 should be moved to a foot note.
Thank you
Page 3 line 26, the reference of SENAMHI should follow same pattern as the other references: SENAMHI (year).
Thank you
Page 4 lines 7-8 should belong to the previous paragraph.
Thank you
Page 4 line 16, please rephrase "Additionally, . . ." to clarify its meaning.

Thank you

Page 5 lines 8-9, move the two-line paragraph to section 2.2.2 since this correspond to a WRF analysis.

Thank you

Page 5 lines 16-19, move the extra content to Table 1. In other words, these lines are not necessary since you have Table 1. This table only needs to be updated with the extra information contained in the abovementioned lines.

Thank you

Page 5 line 28, replace "lake it by" for "lake for".

Thank you

Page 6 line 20, use K instead of deg C.

Thank you

Page 10 line 5, remove period in "24. August".

Thank you

Page 11 lines 1-2, the clause "While. . ." is incomplete.

Thank you

Page 13 line 28, first author is duplicated.

Thank you

Page 13 lines 32-33, reference is duplicated.

Thank you

Page 13 line 35, add title of the reference.

Thank you

Page 15 line 7, URL is duplicated

Thank you

Page 15 lines 28-30, authors and URL are duplicated

Thank you

In all corresponding figures need to mask the Altiplano region for analysis at 850-hPa, as in Sprenger et al. (2012).

Thank you

Use brighter colors in Figures 2 and 5 to clearly identify contours, arrows, and annotations.

Thank you

Use larger fonts in all figures.

Thank you

Provide larger figures (e.g. page width) so it is easier to distinguish thin lines (e.g country limits). Need to add the meaning of the dashed line in Figure 2 and 4. I presume is the contour of Altiplano at certain altitude.

Thank you

Indicate the data source in each figure. For example, it is unclear from the caption in Figure 2 if these are GFS or ERA-Interim data.

Thank you

Figure 4 need larger annotations for A1 and A2. Also, panels e) and f) need an annotation to indicate to which site correspond each.

Thank you

Figure 5 need date and time in panel c). In addition, need the timestamp in all analyses (review previous and following figures) unless they are daily composites, which should be stated in the methodology and figure's caption.

Thank you

Figure 11, why using curly vectors? If no physical explanation is provided then authors should use regular vectors.

Thank you. We will either justify it or change to regular vectors

---

## Author Comment (AC2) · 17 Jan 2020

General Comments This manuscript provides a detailed overview of the synoptic and mesoscale patterns associated with a high impact snowstorm over the Andes of southern Peru and Bolivia in August 2013. The manuscript is very well organized and generally well written, although many paragraphs are quite short and could be expanded/combined. The data, methods, and analyses are all appropriate and conclusions are consistent with the results. The discussion section, however, could be strengthened considerably, as there are only a handful of other papers referenced, and comparison of results with other published literature could be beneficial. Many of the figure panels are very small and difficult to read/interpret and some enlargement could be helpful.

We would like to thank the reviewer for the comments expressed here. We agree that the discussion section can be improved and some extra literature can be included to compare our results to other works. The other reviewers also expressed their concern about the figures and we agree that they need to be updated.

Specific Comments

p. 2, line 2: A number of recent studies (e.g., Romatchske and Houze 2013, Mohr et al. 2014, Endries et al. 2018) have demonstrated that much of the precipitation in the central Andes of Peru and Bolivia is stratiform and not exclusively convective.

This is correct. This information must be included since most of the stratiform precipitation occur during night (Perry et al., 2014, 2017). We will add a sentence about this.

p. 2, line 23: How is the spatial distribution of snowfall clear?

We agree that this has to be developed (or maybe reworded). The revised manuscript will address this.

p. 4, line 16: Incomplete sentence?

We will rephrase this sentence.

p. 7, lines 15-19 and Fig. 6: The grey shading is somewhat confusing as there is a class of >100% and >1.0 m corresponding to grey in Figs. 6a and 6b. Suggest removing this grey class from the legend. The colors for the graduated circles are also a bit hard to interpret but this is partly ameliorated by the graduate size. Adding an inset map for the La Paz vicinity could help? It could be placed in the southern Altiplano/northern Chile?

We thank the reviewer for this remark. As remarked in the reply to the general comments, many figures need to be updated and we will gladly take into account these suggestions.

p. 9, lines 1-10 and Fig. 10: It is not clear what the ending heights (pressure or amsl) for the different panels. Please clarify.

Thank you. We will verify this.

p. 11, lines 17-20: What is the rationale in support of there being daytime and nighttime convection during this event? If no strong evidence then suggest changing "convection" to "precipitation" or "snowfall."

We realize there is not enough evidence to use the word convection. We will modify this sentence.

p. 12, lines 14-15: Again, what evidence is there to support the assertion that there was nighttime convection? Per first comment, recent work has demonstrated that nighttime precipitation across the region is mainly stratiform. Are there any manual or acoustic snowfall observations from Chacaltaya or Zongo in Bolivia?

We tried to connect lake instability with convection. But the reviewer is correct that the evidence is lacking for this claim. We will rephrase this sentence.

Technical Comments

We are thankful to the reviewer for pointing out the technical errors. Since we already plan a good amount of rewriting in the revised manuscript, we are not sure this errors will still be there. However, we will take into account those who will be still present.

p. 3, line 20: Consider replacing "trustful" with "reliable"?
Thank you
p. 4, line 4: s not needed in "conditions" p. 8, line 18: "analysis suggest" should be changed to "analysis suggests"
Thank you
p. 9, line 16: Avoid use of apostrophes (doesn't)
Thank you
p. 10, line 5: Period not needed after 24
Thank you
p. 10, line 7: Change "A colder lake restrict" to "A colder lake restricts"
Thank you
p. 11, line 9: Missing "of" between "relationship" and "PV"?
Thank you
p. 11, line 13: Missing "the" between "for" and "Vuille"?
Thank you
p. 12, line 16: Incomplete sentence
Thank you

---

## Author Comment (AC3) · 17 Jan 2020

General comments:

This paper presents an analysis of a snowstorm event that occurred over the central Andes. The aims are to characterise the synoptic evolution and mesoscale processes leading to the event as well as to determine the specific roles of the orographic features and lake Titicaca. ERA-Interim reanalysis data and observations are used for the synoptic analysis whereas WRF model simulations including sensitivity studies are used to analyse the mesoscale features and the roles of the orography and lake. The authors have synthesised an impressive collection of reanalysis data, model output and observational data. This synthesis yields a two-stage mechanism for the synoptic evolution and information on the factors controlling the mesoscale details of the snowfall regions. The topic appears suitable for NHESS. However, the paper requires fairly substantial polishing before it will be ready for publication. The text is perfectly understandable, but has many small English language errors (some, but not all of which, I've indicated below). Some of the figures also need correcting/improving. In several places I struggled to relate the figures to the associated interpretation in the text. Finally, the novelty of the work, its implications, and its relationship to other studies needs to be made clearer. I was left wondering how much the findings could be generalised to other mountain ranges and cases or were unique to this specific event.

We would like to thank the reviewer for the comments that will help us polish this manuscript. We understand the reviewer's concerns and we accept that many improvements can be done.

Major specific comments:

p6, L7: Here you infer cloud formation and cover from plots of water vapour imagery brightness temperature. I think you need to translate for the reader - i.e. cold brightness temperatures imply high cloud tops. I am also struggling to relate your text to the figures: e.g., you say that on the 22nd there is cloud formation over northern Bolivia but the brightness temperature is relatively warm there. Why do you not instead show the visible image at a suitable time as you do for the analogue case in fig. 5c? Also, why do you not use the same colour scale as in fig. 9d and h which is also GOES-13 data?

The reviewer is correct that there is a lack of consistency between satellite images used in both events, this was mainly due to the mainly nocturnal snowfall for the 2013's event and daytime for 2010. As reviewer 1 suggested, we will explore the use of the Infrared band in order to study cloudiness or stick to water vapor band to asses moisture content. In any case, we agree that more consistency in satellite images is needed and we will modify the figures and text in this direction.

p6, L11: Similarly, here you discuss the total water column displacement and a quasistationary cold front relating to Fig 2. But, what you show in fig. 2 is θe, winds and precipitable water. Please help the reader by relating the features you discuss in the text to the fields plotted and also avoid changing terminology e.g. between total column water to precipitable water.

Thank you for this comment. The reply to the previous comment in terms of consistency in the figures is also valid here. The revised manuscript will also be more consistent in the variables and terminology used.

p6, L20: Here you say that the position of the cold front is characterised by high sea level pressure – yes I can see that there are mean sea level pressure contours that align with the Andes but I'm afraid that I can't work out where the cold fronts are from them and as the projection of this plot (and days shown) differ from that used in fig. 2 it's hard to work out exactly where the cold fronts are. Perhaps you could mark them on the plots?

As pointed by reviewer 1, maybe it is not the best idea to characterize the position of the cold front in terms of sea level pressure level. In the revised manuscript, we will update this figure in order to make it more consistent to other figures with respect to cold front positions (perhaps using $\theta_e$) and modify the text accordingly.

p6, L21: What exactly do you mean by "in phase with Rossby wave trains"? The PV structure on the 15th looks very different to that on the 23rd but your text implies that they are similar. Also it is

the -2 PVU contour which marks the wave train rather than air with PV less than -2PVU (which instead indicates stratospheric air). Also change "portrayed by PVU lower than -2 units" to "portrayed by the -2 PVU surface".

We understand the reviewer's concern with this statement. We believe a better (short) introduction to PV analysis is needed in order to make more sense in this section, more precisely regarding the Fig.3 interpretation. A sentence about PV analysis will be added and its interpretation will be rewritten.

Section 3.1.1: You show several plots of model fields in this section but don't state in the captions where the data is from; presumably it is from ERA-Interim reanalysis. Please be clearer about which plots use ERA-I and which your WRF model runs.

Thank you for this remark. We mainly used ERA-I for large scale circulation analysis and WRF for mesoscale circulation. Perhaps this was not made clear in the captions and we will gladly revisit the concerned figures.

Conclusions: At the end of the discussion and conclusions sections I'm still unclear as to the novel results that have come out of your study. Yes, you have addressed the goals laid out in your introduction (to study this event at both the synoptic and mesoscales and to assess the importance of the orography height and lake). However, it's not clear the extent to which this synoptic evolution and the more local impacts of lakes and the orographic height were already known. Please can you more clearly state the novel contributions of your study.

Thank you for this observation. If we understand correctly, the novelty of the work is not well stablished. If that's the case we will gladly revisit this section in order to address this issue.

Minor specific comments:
Abstract: The abstract describes the work completed and findings well. However, I'm left wondering about the implications of the work - could the authors add a final sentence that answers the "so what" question?

Thank you. Certainly the abstract can be argued by a "so what" question. This is a good suggestion and the revised manuscript will answer it.

General: The text is often written as lots of short paragraphs rather than being grouped together into longer paragraphs on a specific top (see especially the introduction which consists of 9 paragraphs, mostly 2 or 3 sentences long each). I appreciate that this is perhaps the writing style of the authors, but it comes across as note like rather than final text. The text would be easier to read if written as fewer, longer paragraphs.

The main author tries to use short sentences to increase reader's understanding and it is indeed the writing style he is trying to adopt. In the other hand, we realize that many short paragraphs can be hard to read and maybe more work to make them longer (but not too much) is needed.
https://insidegovuk.blog.gov.uk/2014/08/04/sentence-length-why-25-words-is-our-limit/

p5, L13: Change "resolution" to "grid spacing" — the scale of features that a model can resolve is several times (often taken as at least 6 times) the grid spacing.

Thank you for the remark. We will used the suggestion.

p28, table 1: This table doesn't seem to add anything to the details given in the text in section 2.2.2 and could be omitted.

This is a fair point. We will either move the text to Table 1 or remove the table. The reviewed manuscript will be less redundant.

Figure 1: It would help readers for the countries and regions mentioned in the text to be labelled on one of the maps, perhaps best on Fig. 1a.

Figure 2: Is the dashed line indicating where the weak winds are masked? This line encompasses the Andes so is it actually where the winds are weak or instead where the 850 hPa surface is below the ground surface? Also note that the acronym PW isn't defined anywhere.

Fig 4e and f: The symbols and text on these panels are very small and I had to zoom in on the pdf to be able to work out which set of joined symbols corresponded to which date - can you make this clearer please? Also, presumably the larger square and circle refer to some start date with the subsequent symbols 6 hours apart until an end date. Please explain this more clearly in the caption.

Figure 6: The labelling on these figure panels is useful but it's quite hard to read black text on a grey background (and the text is also quite small in places) Also, panels a and d have "July" written on them whereas the caption says that they are for August.

Figure 7 e-h: The y-axis label is wrong as the hours are not UTC (the numbers exceed 24). However, for comparison with the text it would be more useful to have the numbers in UTC. Also, I'm confused by the labelling of night and day on these panels. The caption says that the first red dashed line is 0 UTC on the 23rd and so the space between the 1st and 2nd red dashed line covers 0-12 UTC on the 23rd which according to the text (p7, L31) should be night time, but it is labelled as day. Presumably instead the top of the y-axis is at 0 UTC?

Figure 8: In the text (start of section 3.3.1) it says that the 850 hPa θe gradient is used to indicate the front location which is fine. However, the plot instead shows a single isentrope of θe. A small section of this isentrope is then "highlighted by a blue dashed ellipse" to indicate the front position. I'm afraid that I don't understand how the position of the front can be determined from this small contour section. Nor do I understand how a small highlighted region indicates the location of a large-scale, generally linear feature such as a front.

Figure 9: The font size used for the numbers on the colour bars in this figure particularly are tiny. Please can the text be enlarged. I encourage the authors to look again at the font size used in all of their plots as there are other places too where the font size (particularly on labels) is too small to be easily read in a printed version of the paper. Also, what is the blue shading (particularly over the ocean) and underlying grey shading on panels a-c and e-g? Do they indicate water and topography, respectively? Please label Titicaca lake and La Paz city on one of the plots as these locations are mentioned in the corresponding text.

Figure 11: Please label the axes.

We thank the reviewer for the detailed comments about the figures and their captions. As discussed in the reply to reviewer 1 and in some specific comments of this review, many figures need to be updated. We will try to use as many suggestions (like labeling, colors, inconsistencies, etc) and we will bring up to date this reply once the reviewed manuscript include the updated figures. The main text also will be revised accordingly.

Section 3.2.1.: Here you discuss results from your control WRF run but this discussion would probably be better included in the following section (3.3) where you discuss other results from your WRF control run.

This was done in order to keep the order of Fig. 6. However, it may be a good idea to move the WRF paragraph to section 3.3 and modify the order of Fig.6.

p8, L25: Here it says that the convergence zone over the western Cordillera appears to propagate eastwards during the night. From Fig. 9 though it appears more that the convergence simply weakens during the night.

We agree that Fig. 9 doesn't help too much to back up this sentence. Nevertheless, intermediate time convergence fields can confirm (or reject) this. We will check and modify the text accordingly.

p8, last two paragraphs: The description of "night" in these paragraphs is somewhat inconsistent with the earlier definition of local day as being 12-0 UTC and night 0-12 UTC. The corresponding figure panels from which the nighttime evolution is described are at 18, 0, and 6 UTC. Would it be better to show them at 0, 6, and 12 UTC?

The 18, 0, and 6 UTC order is made for night snowfall chronological reasons. We will revise the text to better express this choice and to make it consistent with the earlier "night" definition.

p6, L16: Here it says that "the lake doesn't seem to exert a significant impact on atmospheric instability". However, Fig, 11d-f shows that θe decreases in height immediately above the lake, particularly on the 23 Aug. Hence this air is potentially unstable.

The reviewer makes a good point here. We were expressing instability in terms of vertical motion, which is incorrect. We will correct this adding a sentence about θe

p10, L11: Please mark CP also on one of the maps in figure 12.

We will update figure 12 using this suggestions

p11, L5: The phrase "high-level PV fields propagated downwards from a trough axis" doesn't make sense. I think you mean that air with high PV values propagated downwards, but I'm also not sure that "propagating" is an appropriate term here. p11, L9: The phrase "The relationship PV streamers/cold surges were ... " is missing a word. Do you mean "The relationship between PV streamers and cold surges were . . . "?

These remarks are helpful and we will use them in the revised manuscript.

Technical errors:

Note that not all of the small English language errors are included here.
Abstract and elsewhere: Usually adjectives should be hyphenated before a noun e.g. "large-scale analyses", "low-level blocking". This seems to have been done a bit randomly in the abstract at least (i.e. sometimes the adjectives are hyphenated and other times they are not).

The changes proposed by the reviewer in the specific comments will imply some rewriting and it is possible that some technical errors will be corrected in the process. However, we will take into account the unchanged errors and we thank the reviewer.

Abstract and p3, L9: Change "2013's" to "2013".
Thank you
p2, first line: change semicolon to a comma.
Thank you
p2, L13: change "circulation" to "circulations".
Thank you
p2, L18: remove comma after gap.
Thank you
p2, L29: change "emergency state" to "state of emergency".
Thank you
p3, L8: change "introdude" to "introduce".
Thank you
p3, L10 and 11: change "contain" to "contains", "summarize" to "summarizes", and "include" to "includes".
Thank you
p3, L15: Change to "For this study about heavy snowfall over complex orography we used the following datasets".
Thank you
p3, L17: Add space after "Andes".
Thank you
p3, L29: Remove brackets around "SENAMHI".
Thank you
p4, L4: Change "conditions" to "condition".
Thank you
p4, L8: Change to "for a spatial assessment of cloud cover".
Thank you

p4, L10: Change "asses" to "assess".
Thank you
p4, L12: Change to "from the surface".
Thank you
p4, L:23 Add space after "processes".
Thank you
p5, L15: Change "runs" to "run".
Thank you
p5, L22: Spelling "lengths".
Thank you
p5, L28: Remove "it".
Thank you
p5, L29: Replace "are described" by "is".
Thank you
p6, L15: Change "correspond" to "corresponds".
Thank you
p6, L26: Change "by" to "on".
Thank you
p7, L26: Change to "we remind the reader that".
Thank you
p8, L28: Change "this" to "these".
Thank you
p9, L5: Change "shows" to "show".
Thank you
p9, L7: Change "origination" to "originating".
Thank you
p9, L9: Change "follows" to "follow.
Thank you
p9, L20: Change "shows" to "show".
Thank you
p10, L5: Remove full stop after "24".
Thank you
p10, L7: Change "restrict" to "restricts".
Thank you
p10, L15: Change "shows" to "show".
Thank you
p10, L16: Change "than" to "to".
Thank you
p11, L13: Add "the" after "for".
Thank you
p11, L23: Change "them" to "it".
Thank you
p11, L26: Change "suggest" to "suggests".
Thank you
p11, L28: Change "confirm" to "confirms".
Thank you
p11, L29: Change "shifted" to "shifting".
Thank you
p12, L3: The punctuation in this paragraph and the following is confused. For example a list following a colon needs to be separated by commas or semicolons
Thank you